# Pan-LUT: Efficient Pan-sharpening via Learnable Look-Up Tables

**Zhongnan Cai**[1][*]  **Yingying Wang**[1][*]  **Hui Zheng**[1]  **Panwang Pan**[2]
**ZiXu Lin**[1]  **Ge Meng**[1]  **Chenxin Li**[3]  **Chunming He**[4]
**Jiaxin Xie**[1]  **Yunlong Lin**[1][†]  **Junbin Lu**[5]  **Yue Huang**[1]  **Xinghao Ding**[1][‡]

[1]Key Laboratory of Multimedia Trusted Perception and Efficient Computing,
Ministry of Education of China, Xiamen University, Xiamen, Fujian, China
[2]ByteDance [3]The Chinese University of Hong Kong
[4]Duke University [5]University of Washington

## Abstract

Recently, deep learning-based pan-sharpening algorithms have achieved notable advancements over traditional methods. However, deep learning-based methods incur substantial computational overhead during inference, especially with large images. This excessive computational demand limits the applicability of these methods in real-world scenarios, particularly in the absence of dedicated computing devices such as GPUs and TPUs. To address these challenges, we propose Pan-LUT, a novel learnable look-up table (LUT) framework for pan-sharpening that strikes a balance between performance and computational efficiency for large remote sensing images. Our method makes it possible to process 15K×15K remote sensing images on a 24GB GPU. To finely control the spectral transformation, we devise the PAN-guided look-up table (PGLUT) for channel-wise spectral mapping. To effectively capture fine-grained spatial details, we introduce the spatial details look-up table (SDLUT). Furthermore, to adaptively aggregate channel information for generating high-resolution multispectral images, we design an adaptive output look-up table (AOLUT). Our model contains fewer than 700K parameters and processes a 9K×9K image in under 1 ms using one RTX 2080 Ti GPU, demonstrating significantly faster performance compared to other methods. Experiments reveal that Pan-LUT efficiently processes large remote sensing images in a lightweight manner, bridging the gap to real-world applications. Furthermore, our model surpasses SOTA methods in full-resolution scenes under real-world conditions, highlighting its effectiveness and efficiency. We also extend our method to general image fusion tasks. The source code is available at
`https://github.com/CZhongnan/Pan-LUT`.

## 1 Introduction

High-resolution multispectral (HRMS) images are widely used in applications such as military operations, environmental monitoring, and mapping. However, due to the limitations of physical sensors, these images are challenging to obtain. Pan-sharpening addresses this issue by fusing high-resolution panchromatic (PAN) images with low-resolution multispectral (LRMS) images, producing high-quality HRMS images through complementary integration [55] [61] [42]. Recently, numerous pan-sharpening methods have been proposed, which can be generally categorized into

---

[*]Equal Contribution.

[†]Project Leader.

[‡]Corresponding author.

39th Conference on Neural Information Processing Systems (NeurIPS 2025).

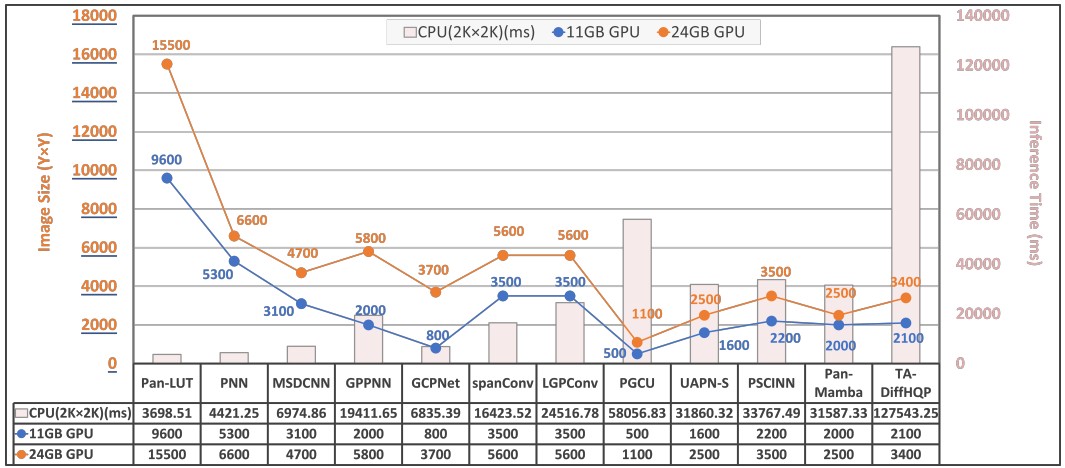

| | Pan-LUT | PNN | MSDCNN | GPPNN | GCPNet | spanConv | LGPConv | PGCU | UAPN-S | PSCINN | Pan-Mamba | TA-DiffHQP |
|---|---|---|---|---|---|---|---|---|---|---|---|---|
| CPU(2K×2K)(ms) | 3698.51 | 4421.25 | 6974.86 | 19411.65 | 6835.39 | 16423.52 | 24516.78 | 58056.83 | 31860.32 | 33767.49 | 31587.33 | 127543.25 |
| 11GB GPU | 9600 | 5300 | 3100 | 2000 | 800 | 3500 | 3500 | 500 | 1600 | 2200 | 2000 | 2100 |
| 24GB GPU | 15500 | 6600 | 4700 | 5800 | 3700 | 5600 | 5600 | 1100 | 2500 | 3500 | 2500 | 3400 |

Figure 1: **Comparisons of computational efficiency.** Our method can process 9K×9K and 15K×15K images on GPUs with 11GB and 24GB memory, respectively. Meanwhile, we observe that (a) DNN-based methods are highly sensitive to the image size, and (b) in the absence of a GPU, they require a considerable amount of time to process images. In the CPU inference time experiments, all methods were conducted on a workstation equipped with an Intel(R) Xeon(R) Gold 6226R CPU.

two main groups: traditional methods and deep learning-based methods. Traditional methods, such as component substitution (CS) [2] [3] [11], multi-resolution analysis (MRA) [7] [19], and variational optimization (VO) [10] [35], often struggle to restore precise spatial or spectral details in HRMS images. In contrast, deep learning-based pan-sharpening methods have demonstrated exceptional fusion capabilities due to the powerful feature extraction ability of deep neural networks (DNNs) [23] [24] [27] [28]. Masi et al. [33] built the Pan-sharpening Neural Network (PNN) model, which first applied CNN to pan-sharpening field, achieving a significant improvement over traditional methods. Following this, researchers have explored more complicated and deeper networks to further promote the performance of pan-sharpening [13] [25] [34] [56]. However, they overlook a critical practical issue: the need for real-time processing of large remote sensing images in real-world applications. As illustrated in Figure 1, we observe two major limitations in DNN-based approaches: (1) they are highly sensitive to the size of the input images, and (2) they rely heavily on dedicated computing devices such as GPUs and TPUs. Increasing GPU memory does not significantly improve the image size these methods can handle and several of these methods demand a significant amount of time to process images in CPU-only environments. And we will demonstrate in the experiments sections that, even with GPU acceleration, most methods still fail to process large remote sensing images in real time. In practical applications, remote sensing images typically exhibit even higher resolutions, posing additional challenges to existing methods in terms of efficiency and scalability. Moreover, simply increasing network depth does not necessarily lead to better performance, as deeper models are harder to train and often suffer from overfitting due to redundant parameters.

To overcome the aforementioned challenges, we propose a novel learnable Look-Up Table (LUT) framework, called Pan-LUT, which achieves a good balance between performance and computational efficiency in pan-sharpening. Specifically, we replace complex DNN operations with learnable LUTs to enable lightweight deployment in practical applications. To finely control the spectral transformation, we devise the PAN-guided look-up table (PGLUT) for channel-wise spectral mapping. To effectively capture fine-grained spatial details, we introduce the spatial details look-up table (SDLUT). To further enable adaptive channel aggregation for high-resolution multispectral image generation, we design the adaptive output look-up table (AOLUT). The Pan-LUT consists of fewer than 700K parameters and can process 9K×9K images in under 1 ms using a single RTX 2080 Ti GPU. Furthermore, our approach outperforms traditional methods by 7 dB, while maintaining a speed comparable to that of conventional techniques, demonstrating superior speed and efficiency compared to existing methods. Our contributions can be summarized as follows:

- We present Pan-LUT, a novel learnable LUT framework that does not incorporate any network structure. This framework is designed to achieve a strong balance between performance and computational efficiency in pan-sharpening high-resolution remote sensing

images. Our method makes it possible to process 15K×15K remote sensing images on one 24GB GPU.

- To finely control the spectral transformation, we devise the PAN-guided look-up table (PGLUT) for channel-wise spectral mapping. To effectively capture fine-grained spatial details and adaptively learn local contexts, we introduce the spatial details look-up table (SDLUT). To further enable adaptive channel aggregation for high-resolution multispectral image generation, we design the adaptive output look-up table (AOLUT).

- To the best of our knowledge, this is the first attempt to introduce LUTs for efficient pan-sharpening. Extensive experiments on different satellite datasets demonstrate the effectiveness and efficiency of Pan-LUT.

## 2 Related Work

### 2.1 Look-Up Table

Look-up Tables (LUT) are particularly useful for functions of multiple variables, as they store pre-computed outputs for all possible input combinations. For example, in a 1D LUT, a single input index is mapped to an output value, often using linear interpolation for indices that fall between pre-stored values. More complex LUTs, such as 3D LUTs, use three independent input variables, which may require advanced interpolation methods like trilinear or tetrahedral interpolation. Due to its portability, various LUT based solutions have been proposed for image enhancement [6] [22] [26] [39] [52]. For instance, Zeng et al. [52] and Wang et al. [39] propose image-adaptive 3D LUTs for efficient single-image enhancement. These approaches rely on a network weight predictor to fuse different 3D LUTs, which may pose a limitation on platforms under resource-constrained conditions. Additionally, LUT-based methods have been explored in the area of super-resolution [18] [21] [31] [29]. SRLUT [18] trains a deep super-resolution (SR) network with a restricted receptive field and then caches the output values from the learned SR network in LUTs. However, issues such as performance degradation arise when large patches are cached in LUTs, prompting the development of strategies like MuLUT [21], which introduces multiple LUT variants and a fine-tuning strategy to improve performance. To further enhance the functionality of LUTs, architectures like SPLUT [31] and RCLUT [29] have been proposed. SPLUT processes different image information separately using multiple LUTs, while RCLUT introduces a plugin module to improve LUT-based models with minimal additional computational cost.

### 2.2 Traditional Pan-sharpening Methods

Traditional fusion techniques encompass component substitution (CS), multi-resolution analysis (MRA), and variational optimization (VO). CS methods, such as IHS [2], Brovey [11], and PCA [3], utilize spatial details from high-resolution panchromatic (PAN) images to replace corresponding details in low-resolution multispectral (LRMS) images, which can lead to spectral distortion due to the incomplete incorporation of spectral information. MRA techniques, including DWT [19] and ATWT [7], apply multi-resolution decomposition to merge PAN and LRMS images, which enables better preservation of spectral information and reduces spectral distortion. VO methods, such as Bayesian [10] and Total Variation [35], formulate the fusion process as an optimization problem by iteratively minimizing the loss function. While VO methods show promising results, they encounter challenges in optimizing model design and loss functions. Although these approaches have yielded certain improvements, their performance remains constrained by the inadequate modeling, which restricts further advancements in pan-sharpening accuracy and quality.

### 2.3 Deep Learning-based Methods

Deep learning-based methods have emerged as the dominant approach for pan-sharpening in recent years [40] [14] [43]. The PNN [33] model, inspired by SRCNN [8], is the first to introduce CNNs into this domain, surpassing traditional methods. Models like PanNet [49] and MSDCNN [50] further enhance performance by leveraging residual connections and multi-scale convolutions, effectively capturing high-frequency details and supporting a wide range of remote sensing applications. Since then, more complex CNN-based architectures [4] [15] [62] have been proposed in this field to improve the mapping ability of pan-sharpening. Models like GPPNN [45], MMNet [47], and ARFNet [46],

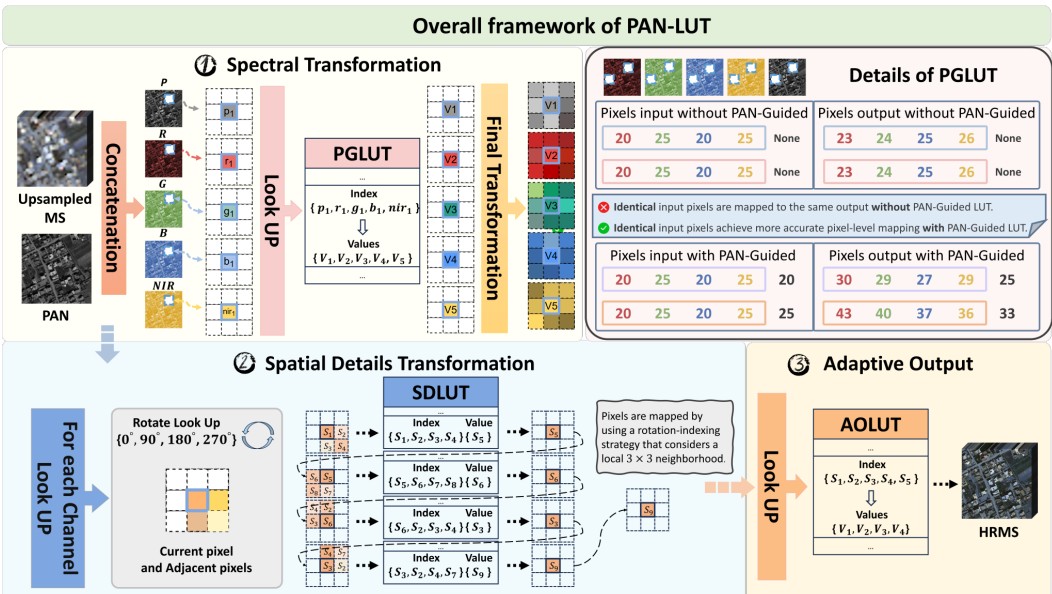

Figure 2: **The overall framework of our proposed Pan-LUT**. PGLUT is a spectral transformation LUT designed to extract spectral information, SDLUT is a spatial detail transformation LUT for capturing texture features, and AOLUT is an adaptive output LUT used to aggregate channel information.

which enhance interpretability through deep unfolding techniques and accelerate model convergence. Spatial adaptive convolution methods, such as LAGConv [17] and CANConv [9], can adaptively generate different convolution kernel parameters based on various spatial locations, enabling them to accommodate different spatial regions. Other approaches, including SFDI [60] and MSDDN [50], utilize Fourier transforms to capture high-frequency features. Transformer-based architectures, such as INN-former [59], Panformer [57] and DRFormer [53] combine CNNs and Transformers to capture both local and global features. Instead of learning a deterministic mapping, generative models such as UCGAN [58], PanFlow [48] and PSCINN [38] generate a distribution of possible outputs for the given inputs. In addition, several studies have investigated lightweight pan-sharpening methods, including SpanConv [5] and LGPConv [54]. However, we show that current lightweight methods are only lightweight in the context of low-resolution images. Despite their promising results, these advanced methods come with high computational costs, limiting their practical applicability.

## 3 Method

Given the PAN image ($P \in R^{H \times W \times 1}$) and the MS image ($MS \in R^{H/r \times W/r \times C}$), pan-sharpening aims to fuse the complementary information to generate the desirable high spatial resolution MS image ($HRMS \in R^{H \times W \times C}$). Here, $H$ and $W$ denote the height and width of the images, $r$ represents the spatial resolution ratio, with a value of 4, and $C$ denotes the number of spectral bands.

### 3.1 Framework

The overall framework of our proposed Pan-LUT is illustrated in Figure 2, which consists of three specifically designed LUTs. (1) To finely control the spectral transformation, we devise the PAN-guided look-up table (PGLUT) for channel-wise spectral mapping, which incorporates a PAN-guided indexing strategy and a pentalinear interpolation technique. (2) To effectively capture fine-grained spatial details and adaptively learn local contexts, we introduce the spatial details look-up table (SDLUT), which incorporates a rotation-enhanced indexing strategy and a quadrilinear interpolation technique. (3) To further enable adaptive channel aggregation for high-resolution multispectral image generation, we design the adaptive output look-up table (AOLUT), which incorporates a PAN-guided indexing strategy and a pentalinear interpolation technique. Specifically, given the PAN image ($P \in \mathbb{R}^{H \times W \times 1}$) and the upsampled MS image ($MS \in \mathbb{R}^{H \times W \times C}$), they are concatenated

into $PM \in \mathbb{R}^{H \times W \times (C+1)}$ and passed through the PGLUT for channel-wise spectral mapping, producing the output $V_{pg} \in \mathbb{R}^{H \times W \times (C+1)}$. SDLUT then takes $V_{pg}$ as input to generate local spatial details, yielding $V_{sd} \in \mathbb{R}^{H \times W \times (C+1)}$. Finally, AOLUT takes $V_{sd}$ as input to generate the final $HRMS \in R^{H \times W \times C}$ result:

$$V_{pg} = PGLUT(PM), V_{sd} = SDLUT(V_{pg}), HRMS = AOLUT(V_{sd}). \tag{1}$$

## 3.2 Spectral Transformation

To preserve the rich spectral information of the MS image, we propose the PAN-guided look-up table (PGLUT), which leverages the PAN image as guidance to finely control the spectral transformation. Specifically, PGLUT is represented as a 5-dimensional matrix containing $N^5$ elements, where $N$ denotes the number of bins per dimension. Each element corresponds to a sampling point, defining a set of indexed input pixels $\{I_{(i,j,k,m,n)}\}_{i,j,k,m,n=0,...,N-1}$ and their corresponding output pixels $\{O_{(i,j,k,m,n)}\}_{i,j,k,m,n=0,...,N-1}$. Here, $I \in \{pa, r, g, b, nir\}$ represents the pixel values from the PAN and MS images, while $O \in \{R, G, B, NIR\}$ denotes the corresponding cached output pixels. Since the LUT elements are discretely distributed in space, the output value cannot be directly retrieved from the LUT. For an input value $\{pa_{(w,h)}^I, r_{(w,h)}^I, g_{(w,h)}^I, b_{(w,h)}^I, nir_{(w,h)}^I\}$, where $(w, h)$ denotes the spatial position of a pixel in the image.

**PAN-guided indexing strategy.** As illustrated in Figure 2, we introduce an indexing strategy for precise spectral transformation, referred to as the PAN-guided indexing strategy. In a MS image, pixels from different spatial locations may have identical values (e.g., $r_i = r_j, g_i = g_j, b_i = b_j, nir_i = nir_j$, where $i \neq j$). The LUT maps these identical inputs to the same output. The PAN-guided indexing strategy provides a more flexible indexing mechanism for the LUT. Specifically, it additionally considers the pixels at corresponding spatial positions in the PAN image (e.g., $r_i = r_j, g_i = g_j, b_i = b_j, nir_i = nir_j, pa_i \neq pa_j$, where $i \neq j$), thereby achieving finer-grained mapping. Specifically, PGLUT first performs a lookup operation to locate the corresponding input pixel in the LUT:

$$x = \frac{pa_{(w,h)}^I}{V_{max}} \cdot N, y = \frac{r_{(w,h)}^I}{V_{max}} \cdot N, z = \frac{g_{(w,h)}^I}{V_{max}} \cdot N,$$
$$s = \frac{b_{(w,h)}^I}{V_{max}} \cdot N, e = \frac{nir_{(w,h)}^I}{V_{max}} \cdot N, \tag{2}$$

where $V_{max}$ denotes the maximum value (e.g., 255, 1023 or 2047). The coordinates of the sampling points, $L = \{(i + c, j + c, k + c, m + c, n + c)\}$, with $c \in \{0, 1\}$, can be derived as follows:

$$i = \lfloor x \rfloor, j = \lfloor y \rfloor, k = \lfloor z \rfloor, m = \lfloor s \rfloor, n = \lfloor e \rfloor, \tag{3}$$

where $\lfloor \cdot \rfloor$ denotes the floor function. $\{\mathbf{d}_l\}_{l=x,y,z,s,e}$ represents the offset of the input index $(x, y, z, s, e)$ relative to the defined sampling point $(i, j, k, m, n)$, e.g., $\mathbf{d}_x = x - i$.

**Pentalinear Interpolation.** After locating 32 adjacent points, an appropriate interpolation technique is applied to these sampled values to generate the output:

$$O_{(x,y,z,s,e)} = PInterpolation(LUT[L], \{\mathbf{d}_l\}), \tag{4}$$

where $PInterpolation(\cdot)$ denotes the pentalinear interpolation. More details about PGLUT and the pentalinear interpolation can be found in Section A.2.

## 3.3 Spatial Details Transformation

PGLUT is essentially a channel-wise 5D LUT that operates globally, which limits its ability to capture local spatial information. To effectively capture fine-grained spatial details and adaptively learn local contexts, we propose the Spatial Details Lookup Table (SDLUT).

**Rotation-indexing strategy.** As illustrated in Figure 2, given a pixel $p_{(w,h)}$, SDLUT processes this pixel along with its neighboring pixels as input. During the training phase, we employ a Rotation-indexing strategy to further expand the receptive field, which can be formulated as:

$$p_{(w,h)}^1 = f_{SDLUT}(p_{(w,h)}, p_{(w+1,h)}, p_{(w,h+1)}, p_{(w+1,h+1)}),$$
$$p_{(w,h)}^2 = f_{SDLUT}(p_{(w,h)}^1, p_{(w+1,h)}^1, p_{(w+1,h-1)}^1, p_{(w,h-1)}^1),$$
$$p_{(w,h)}^3 = f_{SDLUT}(p_{(w,h)}^2, p_{(w+1,h)}^2, p_{(w,h+1)}^2, p_{(w+1,h+1)}^2),$$
$$V_{(w,h)} = f_{SDLUT}(p_{(w,h)}^3, p_{(w+1,h)}^3, p_{(w,h+1)}^3, p_{(w+1,h+1)}^3), \tag{5}$$

Table 1: Quantitative comparison across three satellite datasets. The best outcomes are highlighted in red. ↑ indicates better performance with increasing values, while ↓ signifies improved performance with decreasing values.

| Method | WorldView-II | | | | GaoFen2 | | | | Worldview-III | | | | Param(M) | Inference (ms) | |
|---|---|---|---|---|---|---|---|---|---|---|---|---|---|---|---|
| | PSNR↑ | SSIM↑ | SAM↓ | ERGAS↓ | PSNR↑ | SSIM↑ | SAM↓ | ERGAS↓ | PSNR↑ | SSIM↑ | SAM↓ | ERGAS↓ | | 2K×2K | 4K×4K |
| Brovey | 35.8646 | 0.9216 | 0.0403 | 1.8238 | 37.7974 | 0.9026 | 0.0218 | 1.3720 | 22.5060 | 0.5466 | 0.1159 | 8.2331 | - | 0.28 | 0.33 |
| IHS | 35.2962 | 0.9027 | 0.0461 | 2.0278 | 38.1754 | 0.9100 | 0.0243 | 1.5336 | 22.5579 | 0.5354 | 0.1266 | 8.3616 | - | 0.23 | 0.26 |
| SFIM | 34.1297 | 0.8975 | 0.0439 | 2.3449 | 36.9060 | 0.8882 | 0.0318 | 1.7398 | 21.8212 | 0.5457 | 0.1208 | 8.9730 | - | 0.32 | 0.47 |
| GS | 35.6376 | 0.9176 | 0.0423 | 1.8774 | 37.2260 | 0.9034 | 0.0309 | 1.6736 | 22.5608 | 0.5470 | 0.1217 | 8.2433 | - | 0.75 | 0.87 |
| PNN | 40.7550 | 0.9624 | 0.0259 | 1.0646 | 43.1208 | 0.9704 | 0.0172 | 0.8528 | 29.9418 | 0.9121 | 0.0824 | 3.3206 | 0.0689 | 12.81 | 54.59 |
| PanNet | 40.8176 | 0.9626 | 0.0257 | 1.0557 | 43.0659 | 0.9685 | 0.0178 | 0.8577 | 29.6840 | 0.9072 | 0.0851 | 3.4263 | 0.0688 | 29.52 | OOM |
| MSDCNN | 41.3355 | 0.9664 | 0.0242 | 0.9940 | 45.6847 | 0.9827 | 0.0135 | 0.6389 | 30.3038 | 0.9184 | 0.0782 | 3.1884 | 0.2390 | 49.82 | OOM |
| Pan-GAN | 39.1025 | 0.9562 | 0.0303 | 1.2954 | 41.4468 | 0.9661 | 0.0205 | 1.0593 | 28.4959 | 0.8897 | 0.0998 | 3.9067 | 0.0915 | 40.83 | OOM |
| GPPNN | 41.1622 | 0.9684 | 0.0244 | 1.0315 | 44.2145 | 0.9815 | 0.0137 | 0.7361 | 30.1785 | 0.9175 | 0.0776 | 3.2593 | 0.1198 | 43.60 | OOM |
| SFDI | 41.7244 | 0.9725 | 0.0220 | 0.9506 | 47.4712 | 0.9901 | 0.0102 | 0.5462 | 30.5971 | 0.9236 | 0.0741 | 3.0798 | 0.0871 | 65.37 | OOM |
| UCGAN | 40.0545 | 0.9553 | 0.0275 | 1.1734 | 42.3634 | 0.9557 | 0.0194 | 0.9480 | 28.6705 | 0.8851 | 0.0990 | 3.8696 | 0.2109 | 52.06 | OOM |
| PanFlow | 41.8584 | 0.9712 | 0.0224 | 0.9335 | 47.2533 | 0.9884 | 0.0103 | 0.5512 | 30.4873 | 0.9221 | 0.0751 | 3.1142 | 0.0873 | 54.19 | OOM |
| PSCINN | 41.8520 | 0.9703 | 0.0223 | 0.9407 | 47.1100 | 0.9878 | 0.0107 | 0.5612 | 30.5599 | 0.9230 | 0.0748 | 3.1033 | 3.3209 | 60.74 | OOM |
| Pan-Mamba | 42.2354 | 0.9729 | 0.0212 | 0.8975 | 47.6453 | 0.9894 | 0.0103 | 0.5286 | 31.1551 | 0.9299 | 0.0702 | 2.8942 | 0.1827 | 87.74 | OOM |
| TA-DiffHQP | 42.1255 | 0.9752 | 0.0211 | 0.9023 | 47.7716 | 0.9900 | 0.0101 | 0.5378 | 31.3369 | 0.9302 | 0.0737 | 2.6369 | 2.6000 | 998.58 | OOM |
| Pan-LUT (**Ours**) | 40.8555 | 0.9633 | 0.0254 | 1.0339 | 43.7466 | 0.9726 | 0.0169 | 0.8027 | 29.7376 | 0.9106 | 0.0815 | 3.3934 | 0.6626 | **0.38** | **0.54** |

where $f_{SDLUT}(\cdot)$ denotes the lookup and interpolation process in the LUT retrieval. More details about SDLUT and the quadrilinear interpolation can be found in Section A.3.

## 3.4 Adaptive Output

For the feature channel pixels from the SDLUT $(V^1_{(w,h)}, V^2_{(w,h)}, V^3_{(w,h)}, V^4_{(w,h)}, V^5_{(w,h)})$, AOLUT adaptively aggregates channel information to generate high-resolution multispectral channel pixels $\{R^O_{(w,h)}, G^O_{(w,h)}, B^O_{(w,h)}, NIR^O_{(w,h)}\}$. Pixel-level Transformation can be formulated as:

$$\{R^O_{(w,h)}, G^O_{(w,h)}, B^O_{(w,h)}, NIR^O_{(w,h)}\} = f_{AOLUT}(V^1_{(w,h)}, V^2_{(w,h)}, V^3_{(w,h)}, V^4_{(w,h)}, V^5_{(w,h)}), \quad (6)$$

where $f_{AOLUT}(\cdot)$ denotes the lookup and interpolation process in the LUT retrieval. More details about AOLUT and the pentalinear interpolation can be found in Section A.4.

## 3.5 Loss Function

To achieve satisfying pan-sharpening results, we propose a joint loss for network training. Suppose the batch size is $T$. We first utilize the MSE loss:

$$\mathcal{L}_{mse} = \frac{1}{T} \sum_{t=1}^{T} \|HRMS_t - GT_t\|^2, \quad (7)$$

where $HRMS$ and $GT$ denote the network output and the corresponding ground truth, respectively.

To enhance the stability and robustness of the learned LUTs, we incorporate smoothness regularization $\mathcal{L}_s$ and monotonicity regularization $\mathcal{L}_m$:

$$\mathcal{L}_s = \mathcal{L}_s^{PG} + \mathcal{L}_s^{SD} + \mathcal{L}_s^{AO}, \mathcal{L}_m = \mathcal{L}_m^{PG} + \mathcal{L}_m^{SD} + \mathcal{L}_m^{AO}, \quad (8)$$

where $\mathcal{L}_s^{PG}, \mathcal{L}_s^{SD}$, and $\mathcal{L}_s^{AO}$ denote the smoothness regularizations for PGLUT, SDLUT, and AOLUT, while $\mathcal{L}_m^{PG}, \mathcal{L}_m^{SD}$, and $\mathcal{L}_m^{AO}$ represent the monotonicity regularizations for PGLUT, SDLUT, and AOLUT, respectively.

Taking SDLUT as an example, the smoothness regularization can be defined as:

$$\begin{aligned}
\mathcal{L}_s^{SD} = \sum_{O \in \{l,o,c,a\}} \sum_{i,j,k,m=0}^{N-1} (&\left\|O_{(i+1,j,k,m)} - O_{(i,j,k,m)}\right\|^2 \\
&+ \left\|O_{(i,j+1,k,m)} - O_{(i,j,k,m)}\right\|^2 \\
&+ \left\|O_{(i,j,k+1,m)} - O_{(i,j,k,m)}\right\|^2 \\
&+ \left\|O_{(i,j,k,m+1)} - O_{(i,j,k,m)}\right\|^2),
\end{aligned} \quad (9)$$

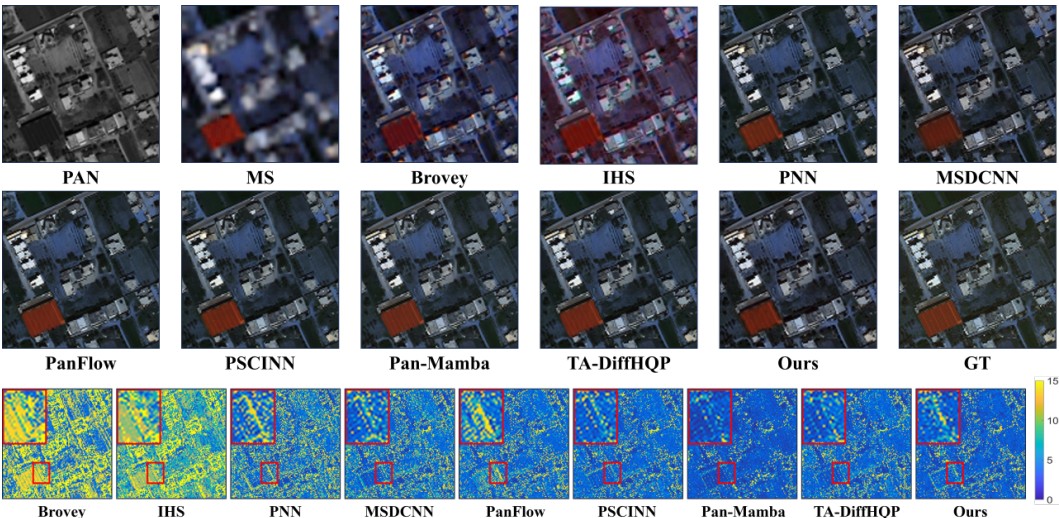

Figure 3: Visual comparison on WorldView-III dataset. The last row visualizes the MSE residues between the pan-sharpening results and the ground truth.

where $N$ represents the number of bins in each dimension of the LUT. $O_{(i,j,k,m)}$ is the corresponding output for the defined sampling point $(i, j, k, m)$ in LUT. The definitions of $\mathcal{L}s^{PG}$ and $\mathcal{L}s^{AO}$ are similar to those in Equation 9.

The monotonicity regularization in AOLUT can be defined as:

$$
\begin{aligned}
\mathcal{L}_m^{SD} = \sum_{O \in \{l,o,c,a\}} \sum_{i,j,k,m=0}^{N-1} & [g(O_{(i,j,k,m)} - O_{(i+1,j,k,m)}) \\
& + g(O_{(i,j,k,m)} - O_{(i,j+1,k,m)}) \\
& + g(O_{(i,j,k,m)} - O_{(i,j,k+1,m)}) \\
& + g(O_{(i,j,k,m)} - O_{(i,j,k,m+1)})],
\end{aligned}
\tag{10}
$$

where $g(\cdot)$ denotes the ReLU activation function. Similarly, $\mathcal{L}_m^{PG}$ and $\mathcal{L}_m^{AO}$ are defined in the same way as in Equation 10.

The final loss functions are as follows:

$$
\mathcal{L} = \mathcal{L}_1 + \lambda_s \mathcal{L}_s + \lambda_m \mathcal{L}_m,
\tag{11}
$$

where the two constant parameters $\lambda_s$ and $\lambda_m$ are used to control the effects of the smoothness and monotonicity regularization terms, respectively. In our experiments, we empirically set $\lambda_s = 0.0001$ and $\lambda_m = 10$. More details about the loss functions can be found in Section A.5

## 4 Experiments

### 4.1 Datasets

Remote sensing datasets from three satellites are used in our experiments, including WorldView-II (WV2), GaoFen2 (GF2) and WorldView-III (WV3). Due to the absence of high-resolution multispectral ground truth images in these datasets, we generate the training set using the Wald protocol tool [37]. Specifically, given the original MS image and its corresponding high-resolution PAN image, they are downsampled by a factor of $r$ to obtain image pairs of MS and PAN, with $r$ set to 4. During training, the original high-resolution MS image is treated as the ground truth, while the MS and PAN images serve as the input image pairs.

### 4.2 Implementation Details

We compare the proposed Pan-LUT model against several pan-sharpening methods on reduced-resolution scenes from WV2, WV3, and GF2 datasets. Specifically, we choose four traditional

Table 2: Evaluation on the real-world full-resolution scenes from WorldView-II dataset. The best values are highlighted by red. The up or down arrow indicates higher or lower metric corresponding to better results.

| Metrics | Brovey | IHS | PNN | MSDCNN | GPPNN | SFDI | UCGAN | PanFlow | PSCINN | Pan-Mamba | TA-DiffHQP | Ours |
|---|---|---|---|---|---|---|---|---|---|---|---|---|
| $D_\lambda \downarrow$ | 0.1026 | 0.1110 | 0.1057 | 0.1063 | 0.0987 | 0.1034 | 0.1042 | 0.0966 | 0.0967 | 0.0966 | 0.0953 | 0.0571 |
| $D_S \downarrow$ | 0.1409 | 0.1556 | 0.1446 | 0.1443 | 0.1312 | 0.1305 | 0.1476 | 0.1274 | 0.1271 | 0.1272 | 0.1129 | 0.0640 |
| QNR↑ | 0.7728 | 0.7527 | 0.7684 | 0.7683 | 0.7859 | 0.7827 | 0.7650 | 0.7910 | 0.7904 | 0.7911 | 0.8025 | 0.8829 |

Figure 4: Visual comparison on the real full-resolution scenes from the WorldView-II dataset. For a more detailed examination of the results, we zoomed-in view on specific parts of the images.

pan-sharpening techniques: Brovey [11], IHS [2], SFIM [30] and GS [20], along with ten deep learning-based approaches: PNN [33], PanNet [49], MSDCNN [50], Pan-GAN [32], SFDI [60], UCGAN [58], PanFlow [48], PSCINN [38], Pan-Mamba [12] and TA-DiffHQP [41]. Several widely used image quality assessment metrics are employed to evaluate the performance of the algorithm, including peak signal-to-noise ratio (PSNR) [16], structural similarity index (SSIM) [44], spectral angle mapper (SAM) [51], relative dimensionless global error in synthesis (ERGAS) [36], spectral distortion index($D_\lambda$), spatial distortion index ($D_S$) and the quality with no reference (QNR) [1]. The PyTorch framework is implemented in our experiment. During the training phase, we employ an ADAM optimizer with $\beta_1 = 0.9$ and $\beta_2 = 0.999$, to update the network parameters for 1000 epochs with a batch size of 1. The learning rate is initialized with $5 \times 10^{-4}$. In parallel, a StepLR learning rate adjustment strategy is employed to reduce the learning rate by half after every 200 iterations. The sizes of PGLUT, SDLUT and AOLUT are set to 9, 9 and 9, respectively.

## 4.3 Comparison with Other Methods

**Evaluation on Reduced-resolution Scene.** The quantitative results across three datasets are presented in Table 1, with the best results highlighted in red. Compared to traditional methods, Pan-LUT achieves an average PSNR improvement of 5dB, 7dB, and 7dB across the three datasets, while maintaining inference speeds comparable to those of traditional methods. It is worth noting that the proposed Pan-LUT does not incorporate any network structure, yet it outperforms some DNN-based methods, such as PanNet and PNN, in terms of both performance and inference time. We also provide visual comparisons for the WV3 datasets, as shown in Figure 3.

**Evaluation on Full-resolution Scene.** To assess the performance and generalization capability of our method on full-resolution scenes under real-world conditions, we first trained Pan-LUT on the reduced-resolution WorldView-II data and then tested it on unseen full-resolution WorldView-II satellite datasets. The real-world dataset consists of 200 newly collected samples from the WorldView-II satellite for evaluation. The results are presented in Table 2. On reduced-resolution scenes, our method falls short of most DNN-based approaches in terms of performance. However, on full-resolution scenes, it outperforms all of them when considering the metrics of $D_\lambda$, $D_S$, and QNR. This demonstrates its strong generalization ability in real-world situations. Additionally, we provide a visual comparison against both traditional and DNN-based methods, as shown in Figure 4.

**Computation Efficiency Comparison.** We conduct three experiments to comprehensively evaluate the computational efficiency of all methods: (1) testing the maximum image size that each method can handle on 11GB and 24GB GPUs; (2) measuring their inference time on the CPU; and (3) evaluating their inference time on an RTX 2080 Ti GPU with 2K×2K and 4K×4K images. For each method, we record the average inference time on 100 images. As shown in Figure 1, even

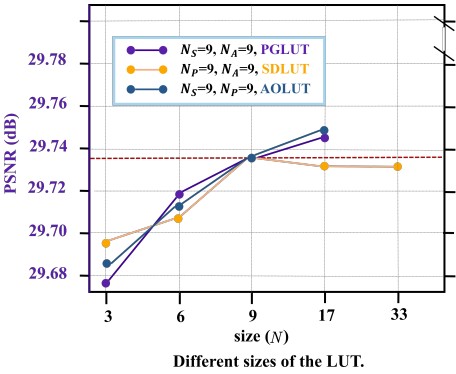

Figure 5: Ablation studies on different sizes of the PGLUT, SDLUT and AOLUT on the WorldView-III dataset.

Table 3: Memory requirements (K).

| Module | 3 | 6 | 9 | 17 | 33 |
|---|---|---|---|---|---|
| AOLUT | 0.97 | 31.10 | 236.20 | 5679.43 | 156541.57 |
| SDLUT | 0.08 | 1.30 | 6.56 | 83.52 | 1185.92 |
| PGLUT | 1.21 | 38.88 | 295.24 | 7099.28 | 195676.96 |

Table 4: Ablation study of PGLUT, SDLUT and AOLUT on the WorldView-III dataset.

| Config | PSNR↑ | SSIM↑ | SAM↓ | ERGAS↓ |
|---|---|---|---|---|
| (i) only PGLUT | 25.2788 | 0.7742 | 0.1130 | 5.4852 |
| (ii) only SDLUT | 26.1433 | 0.8165 | 0.1123 | 5.4211 |
| (iii) only AOLUT | 28.9554 | 0.8677 | 0.1001 | 3.7661 |
| (iv) PGLUT and SDLUT | 29.1315 | 0.8795 | 0.0988 | 3.6785 |
| (v) SDLUT and AOLUT | 29.2754 | 0.9010 | 0.0934 | 3.6033 |
| (vi) PGLUT and AOLUT | 29.4875 | 0.8925 | 0.0900 | 3.4977 |
| ⋆ **Pan-LUT (Ours)** | 29.7376 | 0.9106 | 0.0815 | 3.3934 |

with a 24GB GPU, existing methods fail to process 8K×8K images, while our method can handle 9K×9K images on an 11GB GPU. In environments without GPU acceleration, most methods exhibit unsatisfactory inference speed. As shown in Table 1, Pan-LUT efficiently processes images at all resolutions. Compared to DNN-based methods, it achieves significantly faster inference speeds, while maintaining comparable speed to traditional methods. Our method easily meets the real-time processing requirements on GPUs, outperforming all other methods by a substantial margin. Notably, only PNN is capable of handling remote sensing satellite images at the 4K×4K resolution, highlighting the superior efficiency of our approach.

## 4.4 Ablation Study

**Size of Look-Up Tables.** As shown in Figure 5, changing the LUT size does not lead to a significant drop in performance. This observation suggests that the effectiveness of our proposed method is not dependent on consuming extensive storage resources to increase the LUT size. First, we examine the effect of PGLUT size, denoted $N_P$. Performance improves with larger $N_P$, reaching an optimal point at values $N_P = 9$. Beyond this (from 9 to 17), only a minor gain of 0.01 dB is observed, while the number of parameters increases substantially from 236K to 5M, indicating capacity redundancy. Therefore, we set $N_P = 9$ as the default to balance performance with storage requirements. For SDLUT, denoted $N_S$, increasing $N_S$ from 3 to 9 improves performance, but values above 9 cause a slight performance drop. Similarly, enlarging the AOLUT size $N_A$ yields only minor gains but substantially increases parameters, especially beyond $N_A = 9$. We thus set $N_A = 9$ to balance performance with computational efficiency. We provide the parameter count for each LUT of different sizes in Table 3, which can be calculated as follows:

$$Param_{PGLUT} = 5N^5, Param_{SDLUT} = N^4, Param_{AOLUT} = 4N^5. \tag{12}$$

**Effectiveness of Each LUT.** We further conduct ablation studies to verify the effectiveness of each LUT. Results are listed in Table 4. Our observations are as follows: 1) Comparing (i) with (iv) and (iii) with (v), SDLUT effectively captures fine-grained spatial details from the PAN image, thereby enhancing overall performance. 2) Comparing (ii) with (iv) and (iii) with (iv), PGLUT provides finer control over spectral transformation, resulting in improved performance. 3) Comparing (ii) with (v) and (i) with (vi), AOLUT demonstrates adaptive aggregation capabilities.

## 5 Conclusion

In this paper, we propose a novel learnable LUT framework, called Pan-LUT, which strikes an optimal balance between performance and computational efficiency for high-resolution remote sensing images in pan-sharpening. The proposed method makes it possible to process $15K \times 15K$ remote sensing images on a 24GB GPU and processes a 9K×9K image in under 1 ms using one RTX 2080 Ti GPU. Extensive experiments on various satellite datasets demonstrate the effectiveness and efficiency of Pan-LUT.

# 6  Acknowledgments

This work was supported in part by the National Natural Science Foundation of China under Grant 82172073 and Grant 62271430, and in part by the Open Fund of the National Key Laboratory of Infrared Detection Technologies.

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

# A Appendix

The following contents are provided in the appendix:

- Limitation and Broader Impact.

- More Technical Details.

- More ablation study.

- More comparison experiments.

- More visualization about our experiments.

## A.1 Limitation and Broader Impact

**Limitation.** One limitation of existing methods is the lack of neural network integration. While combining LUTs with neural structures may offer greater modeling capacity, it often comes at the cost of computational efficiency. Conversely, LUTs alone require substantial storage space, making them less suitable for high-dimensional input processing. The space complexity of an n-dimensional LUT is $O(ND^n)$, meaning its size increases exponentially with higher dimensions. This exponential growth severely limits its practical applicability, particularly when dealing with high-dimensional inputs. Lightweight network architectures must be further explored to facilitate efficient integration with LUTs. Meanwhile, more efficient LUT designs are essential for processing high-dimensional inputs.

## A.2 Details of PGLUT

**Input:** Suppose that the MS image ($MS \in R^{H/r \times W/r \times C}$) and the PAN image ($P \in R^{H \times W \times 1}$) are given, the input of PGLUT ($I_{PG} \in R^{H \times W \times (C+1)}$) can be formulated as:

$$I_{PG} = Concat(P, MS \uparrow), \tag{13}$$

where $Concat(\cdot)$ is the concatenation operation. $MS \uparrow$ denotes the upsampled MS image. $MS \uparrow$ and $P$ have the same spatial resolution.

**Output:** the output of PGLUT ($O_{PG} \in R^{H \times W \times (C+1)}$) can be formulated as:

$$O_{PG} = F_{PGLUT}(I_{PG}), \tag{14}$$

where $F_{PGLUT}(\cdot)$ denotes the lookup and pentalinear interpolation based on the PGLUT.

**PAN-guided indexing strategy.** In a MS image, pixels from different spatial locations may have identical values (e.g., $r_i = r_j, g_i = g_j, b_i = b_j, nir_i = nir_j$, where $i \neq j$). The LUT maps these identical inputs to the same output. The PAN-guided indexing strategy provides a more flexible indexing mechanism for the LUT. Specifically, it additionally considers the pixels at corresponding spatial positions in the PAN image (e.g., $r_i = r_j, g_i = g_j, b_i = b_j, nir_i = nir_j, pa_i \neq pa_j$, where $i \neq j$), thereby achieving finer-grained mapping. Given an input value $I_{(w,h)} = \{pa^I_{(w,h)}, r^I_{(w,h)}, g^I_{(w,h)}, b^I_{(w,h)}, nir^I_{(w,h)}\}$, where $(w, h)$ denotes the spatial position of a pixel in the image, the input index to the PGLUT based on the input value can be represented as (x, y, z, s, e). PGLUT first performs a lookup operation to locate the nearest 32 adjacent elements around the input index in the PGLUT. We use $(i, j, k, m, n)$ to denote the coordinates of a defined sampling point in PGLUT, which can be calculated as follows:

$$x = \frac{pa^I_{(w,h)}}{V_{max}} \cdot N, y = \frac{r^I_{(w,h)}}{V_{max}} \cdot N, z = \frac{g^I_{(w,h)}}{V_{max}} \cdot N,$$
$$s = \frac{b^I_{(w,h)}}{V_{max}} \cdot N, e = \frac{nir^I_{(w,h)}}{V_{max}} \cdot N, \tag{15}$$
$$i = \lfloor x \rfloor, j = \lfloor y \rfloor, k = \lfloor z \rfloor, m = \lfloor s \rfloor, n = \lfloor e \rfloor,$$

where $V_{max}$ denotes the maximum value (e.g., 255, 1023 or 2047). $\lfloor \cdot \rfloor$ denotes the floor function. Then, the offset between the input precise index $(x, y, z, s, e)$ and the computed sampling point $(i, j, k, m, n)$ can be computed:

$$d_x = x - i, d_y = y - j, d_z = z - k,$$
$$d_s = s - m, d_e = e - m,$$
$$d_{-x} = 1 - d_x, d_{-y} = 1 - d_y, d_{-z} = 1 - d_z, \tag{16}$$
$$d_{-s} = 1 - d_s, d_{-e} = 1 - d_e.$$

**Pentalinear Interpolation.** After locating 32 adjacent points, an appropriate interpolation technique is applied to generate the output value using the values of these sampled points:

$$
\begin{aligned}
O_{PG(x,y,z,s,e)} = {}& d_{-x}d_{-y}d_{-z}d_{-s}d_{-e}O_{(i,j,k,m,n)} + d_x d_{-y}d_{-z}d_{-s}d_{-e}O_{(i+1,j,k,m,n)} \\
&+ d_{-x}d_y d_{-z}d_{-s}d_{-e}O_{(i,j+1,k,m,n)} + d_{-x}d_{-y}d_z d_{-s}d_{-e}O_{(i,j,k+1,m,n)} \\
&+ d_{-x}d_{-y}d_{-z}d_s d_{-e}O_{(i,j,k,m+1,n)} + d_{-x}d_{-y}d_{-z}d_{-s}d_e O_{(i,j,k,m,n+1)} \\
&+ d_x d_y d_{-z}d_{-s}d_{-e}O_{(i+1,j+1,k,m,n)} + d_x d_{-y}d_z d_{-s}d_{-e}O_{(i+1,j,k+1,m,n)} \\
&+ d_x d_{-y}d_{-z}d_s d_{-e}O_{(i+1,j,k,m+1,n)} + d_x d_{-y}d_{-z}d_{-s}d_e O_{(i+1,j,k,m,n+1)} \\
&+ d_{-x}d_y d_z d_{-s}d_{-e}O_{(i,j+1,k+1,m,n)} + d_{-x}d_y d_{-z}d_s d_{-e}O_{(i,j+1,k,m+1,n)} \\
&+ d_{-x}d_y d_{-z}d_{-s}d_e O_{(i,j+1,k,m,n+1)} + d_{-x}d_{-y}d_z d_s d_{-e}O_{(i,j,k+1,m+1,n)} \\
&+ d_{-x}d_{-y}d_z d_{-s}d_e O_{(i,j,k+1,m,n+1)} + d_{-x}d_{-y}d_{-z}d_s d_e O_{(i,j,k,m+1,n+1)} \\
&+ d_x d_y d_z d_{-s}d_{-e}O_{(i+1,j+1,k+1,m,n)} + d_x d_y d_{-z}d_s d_{-e}O_{(i+1,j+1,k,m+1,n)} \\
&+ d_x d_y d_{-z}d_{-s}d_e O_{(i+1,j+1,k,m,n+1)} + d_x d_{-y}d_z d_s d_{-e}O_{(i+1,j,k+1,m+1,n)} \\
&+ d_x d_{-y}d_z d_{-s}d_e O_{(i+1,j,k+1,m,n+1)} + d_x d_{-y}d_{-z}d_s d_e O_{(i+1,j,k,m+1,n+1)} \\
&+ d_{-x}d_y d_z d_s d_{-e}O_{(i,j+1,k+1,m+1,n)} + d_{-x}d_y d_z d_{-s}d_e O_{(i,j+1,k+1,m,n+1)} \\
&+ d_{-x}d_y d_{-z}d_s d_e O_{(i,j+1,k,m+1,n+1)} + d_{-x}d_{-y}d_z d_s d_e O_{(i,j,k+1,m+1,n+1)} \\
&+ d_x d_y d_z d_s d_{-e}O_{(i+1,j+1,k+1,m+1,n)} + d_x d_y d_z d_{-s}d_e O_{(i+1,j+1,k+1,m,n+1)} \\
&+ d_x d_y d_{-z}d_s d_e O_{(i+1,j+1,k,m+1,n+1)} + d_x d_{-y}d_z d_s d_e O_{(i+1,j,k+1,m+1,n+1)} \\
&+ d_{-x}d_y d_z d_s d_e O_{(i,j+1,k+1,m+1,n+1)} + d_x d_y d_z d_s d_e O_{(i+1,j+1,k+1,m+1,n+1)},
\end{aligned}
\tag{17}
$$

where $O_{(i,j,k,m,n)}$ represents the value of the LUT at the coordinate $(i,j,k,m,n)$.

## A.3  Details of SDLUT

**Input:** The input image ($O_{PG} \in R^{H \times W \times (C+1)}$) of SDLUT is the output of PGLUT.

**Output:** The output of SDLUT ($O_{SD} \in R^{H \times W \times (C+1)}$) can be formulated as:

$$
O_{SD} = F_{SDLUT}(O_{PG}),
\tag{18}
$$

where $F_{SDLUT}(\cdot)$ denotes the lookup and quadrilinear interpolation based on the SDLUT.

**Look Up.** Specifically, for each channel of input, SDLUT operates by iterating through pixels one at a time, treating each as the current pixel during processing. Given the input current pixel and adjacent pixels, which denoted as $p_{(w,h)}$, $p_{(w+1,h)}$, $p_{(w,h+1)}$ and $p_{(w+1,h+1)}$ respectively, the input index to the SDLUT based on the input can be represented as $(x,y,z,s)$. SDLUT first performs a lookup operation to locate the nearest 16 adjacent elements around the input index in the SDLUT. We use $(i,j,k,m)$ to denote the coordinates of a defined sampling point in PGLUT, which can be calculated as follows:

$$
\begin{aligned}
x &= \frac{p_{(w,h)}}{V_{max}} \cdot N, y = \frac{p_{(w+1,h)}}{V_{max}} \cdot N, \\
z &= \frac{p_{(w,h+1)}}{V_{max}} \cdot N, s = \frac{p_{(w+1,h+1)}}{V_{max}} \cdot N, \\
i &= \lfloor x \rfloor, j = \lfloor y \rfloor, k = \lfloor z \rfloor, m = \lfloor s \rfloor,
\end{aligned}
\tag{19}
$$

where $V_{max}$ denotes the maximum pixel value. $\lfloor \cdot \rfloor$ signifies the floor function. Then, the offset between the input precise index $(x,y,z,s)$ and the computed sampling point $(i,j,k,m)$ can be computed:

$$
\begin{aligned}
d_x &= x - i, d_y = y - j, d_z = z - k, d_s = s - m, \\
d_{-x} &= 1 - d_x, d_{-y} = 1 - d_y, \\
d_{-z} &= 1 - d_z, d_{-s} = 1 - d_s.
\end{aligned}
\tag{20}
$$

**Quadrilinear Interpolation.** After locating 16 adjacent points, an appropriate interpolation technique is applied to generate the output value using the values of these sampled points:

$$
\begin{aligned}
O_{SD(x,y,z,s)} = {}& d_{-x}d_{-y}d_{-z}d_{-s}O_{(i,j,k,m)} + d_x d_{-y}d_{-z}d_{-s}O_{(i+1,j,k,m)} \\
&+ d_{-x}d_y d_{-z}d_{-s}O_{(i,j+1,k,m)} + d_{-x}d_{-y}d_z d_{-s}O_{(i,j,k+1,m)} \\
&+ d_{-x}d_{-y}d_{-z}d_s O_{(i,j,k,m+1)} + d_x d_y d_{-z}d_{-s}O_{(i+1,j+1,k,m)} \\
&+ d_x d_{-y}d_z d_{-s}O_{(i+1,j,k+1,m)} + d_x d_{-y}d_{-z}d_s O_{(i+1,j,k,m+1)} \\
&+ d_{-x}d_y d_z d_{-s}O_{(i,j+1,k+1,m)} + d_{-x}d_y d_{-z}d_s O_{(i,j+1,k,m+1)} \\
&+ d_{-x}d_{-y}d_z d_s O_{(i,j,k+1,m+1)} + d_x d_y d_z d_{-s}O_{(i+1,j+1,k+1,m)} \\
&+ d_x d_y d_{-z}d_s O_{(i+1,j+1,k,m+1)} + d_x d_{-y}d_z d_s O_{(i+1,j,k+1,m+1)} \\
&+ d_{-x}d_y d_z d_s O_{(i,j+1,k+1,m+1)} + d_x d_y d_z d_s O_{(i+1,j+1,k+1,m+1)},
\end{aligned}
\tag{21}
$$

where $O_{(i,j,k,m)}$ represents the value of the LUT at the coordinate $(i, j, k, m)$.

**Rotation-indexing strategy.** In general, the performance of the SDLUT can be improved when more pixels are considered. In order to exploit more area in the image, we use a Rotation-indexing strategy in the training phase. For our SDLUT, 4 rotational ensemble with 0, 90, 180, and 270 degrees covers total $3 \times 3$ pixels. Each output from the 4 rotations is defined as follow:

$$
\begin{aligned}
p^1_{(w,h)} &= F_{SDLUT}(p_{(w,h)}, p_{(w+1,h)}, p_{(w,h+1)}, p_{(w+1,h+1)}), \\
p^2_{(w,h)} &= F_{SDLUT}(p^1_{(w,h)}, p^1_{(w+1,h)}, p^1_{(w+1,h-1)}, p^1_{(w,h-1)}), \\
p^3_{(w,h)} &= F_{SDLUT}(p^2_{(w,h)}, p^2_{(w+1,h)}, p^2_{(w,h+1)}, p^2_{(w+1,h+1)}), \\
O_{SD(w,h)} &= F_{SDLUT}(p^3_{(w,h)}, p^3_{(w+1,h)}, p^3_{(w,h+1)}, p^3_{(w+1,h+1)}),
\end{aligned}
\tag{22}
$$

where $F_{SDLUT}(\cdot)$ denotes the lookup and quadrilinear interpolation based on the SDLUT.

**Proof** Let $(w, h)$ denote the pixel located at the w-th column and h-th row of the image. We provide a theoretical proof that Rotation-indexing strategy (RiS) extends the receptive field from $2 \times 2$ to $3 \times 3$. Ideally, we aim to incorporate all pixels within the local $3 \times 3$ region centered at $(w, h)$, corresponding to the input index set:

$$
\begin{aligned}
G_{tgt} = \{&(w-1, h-1), (w, h-1), (w+1, h-1), (w-1, h), \\
&(w, h), (w+1, h), (w-1, h+1), (w, h+1), (w+1, h+1)\}.
\end{aligned}
\tag{23}
$$

By default, SDLUT captures the bottom-right neighborhood of each pixel, with the corresponding input index group defined as:

$$
G_{r0} = \{(w, h), (w+1, h), (w, h+1), (w+1, h+1)\}.
\tag{24}
$$

With the proposed RiS, SDLUT is applied to three rotated versions of the input, thereby effectively capturing a broader set of pixel neighborhoods.

$$
\begin{aligned}
G_{r90} &= \{(w, h), (w-1, h), (w, h+1), (w-1, h+1)\}, \\
G_{r180} &= \{(w, h), (w-1, h), (w, h-1), (w-1, h-1)\}, \\
G_{r270} &= \{(w, h), (w+1, h), (w, h-1), (w+1, h-1)\}.
\end{aligned}
\tag{25}
$$

Then, we can derive the following equation:

$$
G_{r0} \cup G_{r90} \cup G_{r180} \cup G_{r270} = G_{tgt}.
\tag{26}
$$

## A.4 Details of AOLUT

**Input:** The input of AOLUT is the output from the SDLUT ($O_{SD} \in R^{H \times W \times (C+1)}$).

**Output:** the output of AOLUT ($O_{AO} \in R^{H \times W \times C}$) can be formulated as:

$$
O_{AO} = F_{AOLUT}(O_{SD}),
\tag{27}
$$

where $F_{AOLUT}(\cdot)$ denotes the lookup and pentalinear interpolation based on the AOLUT.

Given an input value $I_{(w,h)} = \{O^{c_1}_{SD(w,h)}, O^{c_2}_{SD(w,h)}, O^{c_3}_{SD(w,h)}, O^{c_4}_{SD(w,h)}, O^{c_5}_{SD(w,h)}\}$, where $(w, h)$ denotes the spatial position of a pixel in the image, $\{c_1, c_2, c_3, c_4, c_5\}$ denote the different channels of $O_{SD}$. The input index to the PGLUT based on the input value can be represented as (x, y, z, s, e). PGLUT first performs a lookup operation to locate the nearest 32 adjacent elements around the input index in the PGLUT. We use $(i, j, k, m, n)$ to denote the coordinates of a defined sampling point in PGLUT, which can be calculated as follows:

$$
\begin{aligned}
x &= \frac{O^{c_1}_{SD(w,h)}}{V_{max}} \cdot N, y = \frac{O^{c_2}_{SD(w,h)}}{V_{max}} \cdot N, z = \frac{O^{c_3}_{SD(w,h)}}{V_{max}} \cdot N, \\
s &= \frac{O^{c_4}_{SD(w,h)}}{V_{max}} \cdot N, e = \frac{O^{c_5}_{SD(w,h)}}{V_{max}} \cdot N, \\
i &= \lfloor x \rfloor, j = \lfloor y \rfloor, k = \lfloor z \rfloor, m = \lfloor s \rfloor, n = \lfloor e \rfloor,
\end{aligned}
\tag{28}
$$

where $V_{max}$ denotes the maximum value (e.g., 255, 1023 or 2047). $\lfloor \cdot \rfloor$ denotes the floor function. Then, the offset between the input precise index $(x, y, z, s, e)$ and the computed sampling point $(i, j, k, m, n)$ can be computed:

$$
\begin{aligned}
d_x &= x - i, d_y = y - j, d_z = z - k, \\
d_s &= s - m, d_e = e - m, \\
d_{-x} &= 1 - d_x, d_{-y} = 1 - d_y, d_{-z} = 1 - d_z, \\
d_{-s} &= 1 - d_s, d_{-e} = 1 - d_e.
\end{aligned}
\tag{29}
$$

**Pentalinear Interpolation.** After locating 32 adjacent points, an appropriate interpolation technique is applied to generate the output value using the values of these sampled points:

$$
\begin{aligned}
O_{AO(x,y,z,s,e)} = &\; d_{-x}d_{-y}d_{-z}d_{-s}d_{-e}O_{(i,j,k,m,n)} + d_x d_{-y}d_{-z}d_{-s}d_{-e}O_{(i+1,j,k,m,n)} \\
&+ d_{-x}d_y d_{-z}d_{-s}d_{-e}O_{(i,j+1,k,m,n)} + d_{-x}d_{-y}d_z d_{-s}d_{-e}O_{(i,j,k+1,m,n)} \\
&+ d_{-x}d_{-y}d_{-z}d_s d_{-e}O_{(i,j,k,m+1,n)} + d_{-x}d_{-y}d_{-z}d_{-s}d_e O_{(i,j,k,m,n+1)} \\
&+ d_x d_y d_{-z}d_{-s}d_{-e}O_{(i+1,j+1,k,m,n)} + d_x d_{-y}d_z d_{-s}d_{-e}O_{(i+1,j,k+1,m,n)} \\
&+ d_x d_{-y}d_{-z}d_s d_{-e}O_{(i+1,j,k,m+1,n)} + d_x d_{-y}d_{-z}d_{-s}d_e O_{(i+1,j,k,m,n+1)} \\
&+ d_{-x}d_y d_z d_{-s}d_{-e}O_{(i,j+1,k+1,m,n)} + d_{-x}d_y d_{-z}d_s d_{-e}O_{(i,j+1,k,m+1,n)} \\
&+ d_{-x}d_y d_{-z}d_{-s}d_e O_{(i,j+1,k,m,n+1)} + d_{-x}d_{-y}d_z d_s d_{-e}O_{(i,j,k+1,m+1,n)} \\
&+ d_{-x}d_{-y}d_z d_{-s}d_e O_{(i,j,k+1,m,n+1)} + d_{-x}d_{-y}d_{-z}d_s d_e O_{(i,j,k,m+1,n+1)} \\
&+ d_x d_y d_z d_{-s}d_{-e}O_{(i+1,j+1,k+1,m,n)} + d_x d_y d_{-z}d_s d_{-e}O_{(i+1,j+1,k,m+1,n)} \\
&+ d_x d_y d_{-z}d_{-s}d_e O_{(i+1,j+1,k,m,n+1)} + d_x d_{-y}d_z d_s d_{-e}O_{(i+1,j,k+1,m+1,n)} \\
&+ d_x d_{-y}d_z d_{-s}d_e O_{(i+1,j,k+1,m,n+1)} + d_x d_{-y}d_{-z}d_s d_e O_{(i+1,j,k,m+1,n+1)} \\
&+ d_{-x}d_y d_z d_s d_{-e}O_{(i,j+1,k+1,m+1,n)} + d_{-x}d_y d_z d_{-s}d_e O_{(i,j+1,k+1,m,n+1)} \\
&+ d_{-x}d_y d_{-z}d_s d_e O_{(i,j+1,k,m+1,n+1)} + d_{-x}d_{-y}d_z d_s d_e O_{(i,j,k+1,m+1,n+1)} \\
&+ d_x d_y d_z d_s d_{-e}O_{(i+1,j+1,k+1,m+1,n)} + d_x d_y d_z d_{-s}d_e O_{(i+1,j+1,k+1,m,n+1)} \\
&+ d_x d_y d_{-z}d_s d_e O_{(i+1,j+1,k,m+1,n+1)} + d_x d_{-y}d_z d_s d_e O_{(i+1,j,k+1,m+1,n+1)} \\
&+ d_{-x}d_y d_z d_s d_e O_{(i,j+1,k+1,m+1,n+1)} + d_x d_y d_z d_s d_e O_{(i+1,j+1,k+1,m+1,n+1)},
\end{aligned}
\tag{30}
$$

where $O_{(i,j,k,m,n)}$ represents the value of the LUT at the coordinate $(i,j,k,m,n)$.

## A.5 Details of Loss Function

To achieve satisfying pan-sharpening results, we propose a joint loss for network training. Suppose the batch size is $T$. We first utilize the $\mathcal{L}_1$ loss:

$$
\mathcal{L}_{mse} = \frac{1}{T}\sum_{t=1}^{T}\|HRMS_t - GT_t\|^2,
\tag{31}
$$

where $HRMS$ and $GT$ denote the network output and the corresponding ground truth, respectively.

To enhance the stability and robustness of the learned LUTs, we incorporate smoothness regularization $\mathcal{L}_s$ and monotonicity regularization $\mathcal{L}_m$:

$$
\mathcal{L}_s = \mathcal{L}_s^{PG} + \mathcal{L}_s^{SD} + \mathcal{L}_s^{AO},
\tag{32}
$$

$$
\mathcal{L}_m = \mathcal{L}_m^{PG} + \mathcal{L}_m^{SD} + \mathcal{L}_m^{AO},
\tag{33}
$$

where $\mathcal{L}_s^{PG}$, $\mathcal{L}_s^{SD}$, and $\mathcal{L}_s^{AO}$ denote the smoothness regularizations for PGLUT, SDLUT, and AOLUT, while $\mathcal{L}_m^{PG}$, $\mathcal{L}_m^{SD}$, and $\mathcal{L}_m^{AO}$ represent the monotonicity regularizations for PGLUT, SDLUT, and AOLUT, respectively.

**The Smoothness Regularization of PGLUT and AOLUT**:

$$
\begin{aligned}
\mathcal{L}_s^{PG}, \mathcal{L}_s^{AO} = \sum_{O\in\{l,o,c,a,e\}} \sum_{i,j,k,m,n=0}^{N-1} (&\big\|O_{(i+1,j,k,m,n)} \\
-O_{(i,j,k,m,n)}\big\|^2 + &\big\|O_{(i,j+1,k,m,n)} - O_{(i,j,k,m,n)}\big\|^2 \\
+ &\big\|O_{(i,j,k+1,m,n)} - O_{(i,j,k,m,n)}\big\|^2 \\
+ &\big\|O_{(i,j,k,m+1,n)} - O_{(i,j,k,m,n)}\big\|^2 \\
+ &\big\|O_{(i,j,k,m,n+1)} - O_{(i,j,k,m,n)}\big\|^2).
\end{aligned}
\tag{34}
$$

**The Monotonicity Regularization of PGLUT and AOLUT**:

$$
\begin{aligned}
\mathcal{L}_m^{PG}, \mathcal{L}_m^{AO} = \sum_{O\in\{l,o,c,a,e\}} \sum_{i,j,k,m,n=0}^{N-1} [&g(O_{(i,j,k,m,n)} \\
-O_{(i+1,j,k,m,n)}) + &g(O_{(i,j,k,m,n)} - O_{(i,j+1,k,m,n)}) \\
+ &g(O_{(i,j,k,m,n)} - O_{(i,j,k+1,m,n)}) \\
+ &g(O_{(i,j,k,m,n)} - O_{(i,j,k,m+1,n)}) \\
+ &g(O_{(i,j,k,m,n)} - O_{(i,j,k,m,n+1)})],
\end{aligned}
\tag{35}
$$

Table 5: Ablation study on $\lambda_m$

| Metrics | 0 | 0.1 | 1 | 10 | 100 |
|---|---|---|---|---|---|
| PSNR | 29.5321 | 29.5421 | 29.6241 | **29.7376** | 29.4112 |

Table 6: Ablation study on $\lambda_s$

| Metrics | 0 | 0.00001 | 0.0001 | 0.001 | 0.01 |
|---|---|---|---|---|---|
| PSNR | 29.4746 | 29.4889 | **29.7376** | 29.5423 | 29.5213 |

where $N$ represents the number of bins in each dimension of the LUT. $O_{(i,j,k,m,n)}$ is the corresponding output for the defined sampling point $(i,j,k,m,n)$ in LUT. $g(\cdot)$ denotes the ReLU activation function.

**The Smoothness Regularization of SDLUT**:

$$
\begin{aligned}
\mathcal{L}_s^{SD} = \sum_{O \in \{l,o,c,a\}} \sum_{i,j,k,m=0}^{N-1} (\left\| O_{(i+1,j,k,m)} \right. \\
\left. -O_{(i,j,k,m)} \right\|^2 + \left\| O_{(i,j+1,k,m)} - O_{(i,j,k,m)} \right\|^2 \\
+ \left\| O_{(i,j,k+1,m)} - O_{(i,j,k,m)} \right\|^2 \\
+ \left\| O_{(i,j,k,m+1)} - O_{(i,j,k,m)} \right\|^2),
\end{aligned}
\tag{36}
$$

**The Monotonicity Regularization of SDLUT**:

$$
\begin{aligned}
\mathcal{L}_m^{SD} = \sum_{O \in \{l,o,c,a\}} \sum_{i,j,k,m=0}^{N-1} [g(O_{(i,j,k,m)} \\
-O_{(i+1,j,k,m)}) + g(O_{(i,j,k,m)} - O_{(i,j+1,k,m)}) \\
+g(O_{(i,j,k,m)} - O_{(i,j,k+1,m)}) \\
+g(O_{(i,j,k,m)} - O_{(i,j,k,m+1)})],
\end{aligned}
\tag{37}
$$

where $N$ represents the number of bins in each dimension of the LUT. $O_{(i,j,k,m)}$ is the corresponding output for the defined sampling point $(i,j,k,m)$ in LUT. $g(\cdot)$ denotes the ReLU activation function.

The final loss function is as follows:

$$
\mathcal{L} = \mathcal{L}_1 + \lambda_s \mathcal{L}_s + \lambda_m \mathcal{L}_m,
\tag{38}
$$

where the two constant parameters $\lambda_s$ and $\lambda_m$ are used to control the effects of the smoothness and monotonicity regularization terms, respectively. In our experiments, we empirically set $\lambda_s = 0.0001$ and $\lambda_m = 10$.

## A.6 Selection of regularization parameters.

As shown in Table 5, 6, we vary the $\lambda_s$ and $\lambda_m$ to determine the optimal parameters ($\lambda_m = 10$ and $\lambda_s = 0.0001$). A large $\lambda_s$ (e.g., $> 0.0001$) results in worse PSNR, as the smooth regularization limits the flexibility of LUT transformations. In contrast, the PSNR is insensitive to the choice of $\lambda_m$ since monotonicity is a natural constraint to LUTs.

## A.7 More Comparison With Traditional methods

We further compare our method with additional traditional approaches, as shown in Table 7. While significantly surpassing traditional methods in performance across the three datasets, our approach maintains a comparable processing speed, highlighting its practical efficiency. Notably, some traditional methods are not only time-consuming but also inefficient.

## A.8 More Visualization about Experiments

As shown in Figure 6 and Figure 7, extensive visual comparisons are provided for the WV2 and GF2 datasets. More visual comparisons are provided in the supplementary material.

Table 7: Quantitative comparison across three satellite datasets with traditional methods. The best outcomes are highlighted in red. ↑ indicates better performance with increasing values, while ↓ signifies improved performance with decreasing values.

| Method | WorldView-II | | | | GaoFen2 | | | | Worldview-III | | | | Inference (ms) | |
|---|---|---|---|---|---|---|---|---|---|---|---|---|---|---|
| | PSNR↑ | SSIM↑ | SAM↓ | ERGAS↓ | PSNR↑ | SSIM↑ | SAM↓ | ERGAS↓ | PSNR↑ | SSIM↑ | SAM↓ | ERGAS↓ | $2K \times 2K$ | $4K \times 4K$ |
| PCA | 20.3542 | 0.7002 | 0.3741 | 10.8524 | 19.2933 | 0.7974 | 0.3908 | 14.3228 | 20.4455 | 0.5263 | 0.2693 | 10.3129 | 0.21 | 0.23 |
| Brovey | 35.8646 | 0.9216 | 0.0403 | 1.8238 | 37.7974 | 0.9026 | 0.0218 | 1.3720 | 22.5060 | 0.5466 | 0.1159 | 8.2331 | 0.28 | 0.33 |
| IHS | 35.2962 | 0.9027 | 0.0461 | 2.0278 | 38.1754 | 0.9100 | 0.0243 | 1.5336 | 22.5579 | 0.5354 | 0.1266 | 8.3616 | 0.23 | 0.26 |
| SFIM | 34.1297 | 0.8975 | 0.0439 | 2.3449 | 36.9060 | 0.8882 | 0.0318 | 1.7398 | 21.8212 | 0.5457 | 0.1208 | 8.9730 | 0.32 | 0.47 |
| Wavelet | 34.9827 | 0.8806 | 0.0481 | 2.0907 | 35.7502 | 0.8213 | 0.0283 | 2.0148 | 21.8551 | 0.5216 | 0.1368 | 9.1158 | 13.37 | 21.73 |
| GS | 35.6376 | 0.9176 | 0.0423 | 1.8774 | 37.2260 | 0.9034 | 0.0309 | 1.6736 | 22.5608 | 0.5470 | 0.1217 | 8.2433 | 0.75 | 0.87 |
| GSA | 35.3574 | 0.9219 | 0.097 | 1.7401 | 35.948 | 0.8779 | 0.0368 | 1.9257 | 21.8845 | 0.5458 | 0.1394 | 9.0781 | 0.73 | 0.76 |
| GFPCA | 34.5580 | 0.9038 | 0.0488 | 2.1400 | 37.9443 | 0.9204 | 0.0314 | 1.5604 | 22.3344 | 0.4826 | 0.1294 | 8.3964 | 0.42 | 0.66 |
| PRACS | 34.9671 | 0.9063 | 0.0414 | 1.8725 | 36.2015 | 0.8902 | 0.0372 | 1.8312 | 22.4452 | 0.5535 | 0.1373 | 8.2961 | 2.44 | 4.75 |
| AWLP | 32.2402 | 0.8709 | 0.0457 | 2.4077 | 37.2183 | 0.8917 | 0.0281 | 1.5966 | 21.5792 | 0.5323 | 0.1260 | 9.0636 | 8.39 | 14.17 |
| MTF-GLP-HPM | 31.3946 | 0.8722 | 0.0492 | 3.3040 | 37.9443 | 0.9204 | 0.0314 | 1.5604 | 21.1033 | 0.5505 | 0.1233 | 9.8406 | 4.22 | 6.09 |
| Pan-LUT (**Ours**) | 40.8555 | 0.9633 | 0.0254 | 1.0339 | 43.7466 | 0.9726 | 0.0169 | 0.8027 | 29.7376 | 0.9106 | 0.0815 | 3.3934 | 0.38 | 0.54 |

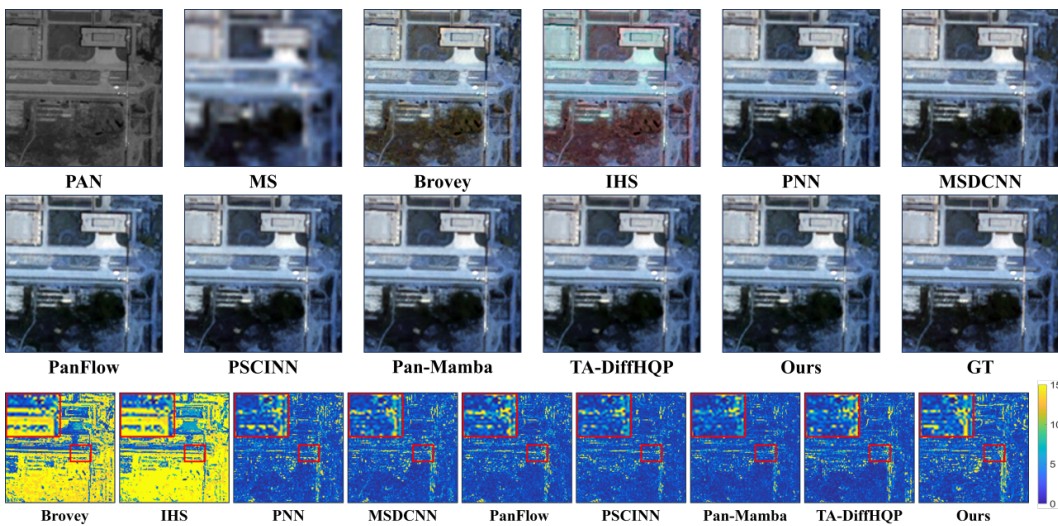

Figure 6: Visual comparison on WV2 dataset. The last row visualizes the MSE residues between the pan-sharpening results and the ground truth.

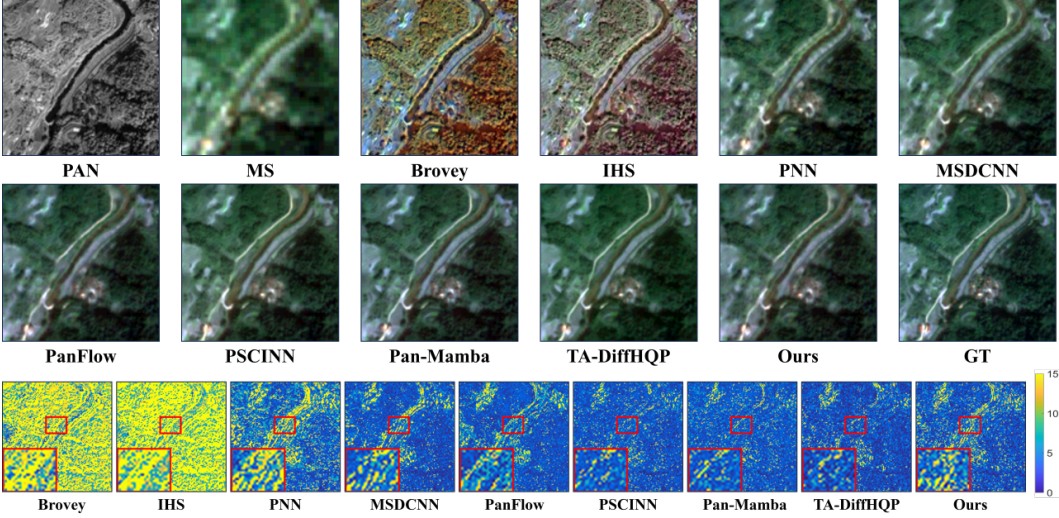

Figure 7: Visual comparison on GaoFen2 dataset. The last row visualizes the MSE residues between the pan-sharpening results and the ground truth.

