# OpenReview forum: "Pan-LUT: Efficient Pan-sharpening via Learnable Look-Up Tables"
_NeurIPS.cc/2025/Conference — NeurIPS 2025 oral_

### Official Review · Reviewer_cahF · 2025-06-10

**Clarity:** 4
**Significance:** 3
**Originality:** 4
**Rating:** 6
**Confidence:** 5

**Summary:**

This paper introduces Pan-LUT, a novel learnable Look-Up Table (LUT) framework designed for pan-sharpening. The core contribution lies in the development of a computationally efficient approach that balances performance with resource constraints, particularly for deployment in scenarios where real-time processing is crucial. The experimental results demonstrate that Pan-LUT achieves competitive performance, particularly in terms of computational efficiency, when compared to existing pan-sharpening methods. Specifically, the proposed method exhibits significantly faster inference times than deep learning-based approaches, while maintaining a reasonable level of accuracy.

**Questions:**

1、The authors should provide a more detailed analysis of the memory complexity. This should include a breakdown of the memory requirement of each component of the proposed method, such as the PGLUT, SDLUT, and AOLUT modules.
2、Further evaluation of additional Pan-LUT variants (e.g., Pan-LUT-4 and Pan-LUT-5) is necessary to better assess the trade-off between performance and computational efficiency.
3、How are the hyperparameters λs and λm​ in Eq (11) set? Here, λs controls the smoothness regularization term, while λm​ controls the monotonicity regularization term in the loss function.

**Ethical Concerns:**

["NO or VERY MINOR ethics concerns only"]

**Final Justification:**

The responses have solved all my concerns，so I decide to raise the final score.

**Limitations:**

N

**Quality:**

4

**Strengths And Weaknesses:**

**Strengths** \
1、The primary strength of this paper lies in its introduction of a novel learnable LUT framework for pan-sharpening.\
2、The proposed framework achieve a practical balance between performance and efficiency, making it a potentially valuable contribution to the field of remote sensing image processing.\
3、Quantitative results demonstrate that the proposed framework substantially outperforms classical approaches. Moreover, it obtains acceptable results with respect to the deep-learning based models, but with a considerably lower processing time.\
**Weaknesses** \
1、The effectiveness of the proposed method, while reasonable, does not consistently achieve state-of-the-art performance when compared to some deep learning-based methods.\
2、The paper lacks illustrative figures that clearly demonstrate the operations of the proposed LUT modules.

---

> ### Author Rebuttal · Authors · 2025-07-26
>
> Thank you for your review.
> # Weakness 1
> Our Pan-LUT is specifically designed to efficiently process high-resolution remote sensing images under resource-constrained conditions. While it does not consistently achieve state-of-the-art performance when compared to some deep learning-based methods, it consistently outperforms other approaches in real-time inference and high-resolution image processing scenarios. Its lightweight architecture enables practical deployment, effectively bridging the gap to real-world applications.
>
> # Weakness 2
> Since it is challenging to visualize operations in 4D or 5D space, we instead provide illustrative figures based on 3D LUTs to clearly demonstrate the structure and operations of the proposed LUT modules. For more details, please refer to the Technical Details section in the supplementary material.
>
> # Question 1
>
> | Module | M=3 | M=6 | M=9 | M=17 | M=33 |
> | :--- | :- | :-- | :-- | :--- | :--- |
> |  | Param (K)  | Param (K)  | Param (K)  | Param (K)  | Param (K)  |
> | AOLUT | 0.97 | 31.10 | 236.20 | 5679.43 | 156541.57 |
> | SDLUT | 0.08 | 1.30 | 6.56 | 83.52 | 1185.92 |
> | PGLUT | 1.21 | 38.88 | 295.24 | 7099.28 | 195676.96 |
>
> The memory complexities of PGLUT, SDLUT, and AOLUT are $O(5M^5)$, $O(4M^4)$, and $O(4M^5)$, respectively, where $M$ denotes the size of the LUT. As shown in Table above and Figure 5 of the manuscript, increasing the LUT size beyond 9 yields only minor performance gains but significantly consumes excessive storage resources. In all experiments, we use $M=9$ as the default setting.
>
> # Question 2
>
> We conduct additional experiments to evaluate more Pan-LUT-N variants, where $N$ denotes the number of stacked PGLUT and SDLUT modules.
>
> | Method | WV2 | GF2 | WV3 | Param (M) | Inference (ms) | Inference (ms) |
> | :---: | :---: | :---: | :---: | :---: | :---: | :---: |
> ||||||$2048\\times2048$|$4096\\times4096$|
> | Pan-LUT-1 | 40.6362 | 43.6559 | 29.6213 | 0.9579 | 0.3812 | 0.5445 |
> | Pan-LUT-2 | 40.8419 | 44.5545 | 29.7929 | 1.3843 | 0.9434 | 1.0824 |
> | Pan-LUT-3 | 41.1362 | 44.8559 | 30.1213 | 1.8108 | 1.4821 | 1.9648 |
> | Pan-LUT-4 | 41.1624 | 44.9756 | 30.1913 | 2.2373 | 2.0621 | 2.7518 |
> | Pan-LUT-5 | 41.2854 | 45.1733 | 30.2471 | 2.6637 | 2.3712 | 2.9915 |
> | Pan-LUT-6 | 41.4025 | 45.3364 | 30.3087 | 3.0902 | 2.7915 | 3.1024 |
>
> As shown, deeper models (e.g., Pan-LUT-4 to Pan-LUT-6) offer only marginal improvements in performance compared to shallower variants (Pan-LUT-1 to Pan-LUT-3), while significantly increasing parameter count. The Pan-LUT-4 already contains over 2 million parameters. From a practical application perspective, we consider Pan-LUT-1, 2, and 3 as the default architectural choices.
>
> # Question 3
>
> | Metrics | $\\lambda_m$=0 | $\\lambda_m$=0.1 | $\\lambda_m$=1 | $\\lambda_m$=10 | $\\lambda_m$=100 |
> | :--- | :--- | :--- | :--- | :--- | :--- |
> | PSNR | 29.4287 | 29.4902 | 29.5275 | **29.6213** | 29.3785 |
>
> | Metrics | $\\lambda_s$=0 | $\\lambda_s$=0.00001 | $\\lambda_s$=0.0001 | $\\lambda_s$=0.001 | $\\lambda_s$=0.01 |
> | :--- | :--- | :--- | :--- | :--- | :--- |
> | PSNR | 29.4163 | 29.4176 | **29.6213** | 29.4137 | 29.4044 |
>
> We tune the hyperparameters $\\lambda_s$ and $\\lambda_m$ to identify the optimal settings, and find that $\\lambda_m = 10$ and $\\lambda_s = 0.0001$ yield the best performance.
> A large $\lambda_s$ (e.g., $>0.0001$) results in worse PSNR, as the smooth regularization limits the flexibility of LUT transformations. In contrast, the PSNR is insensitive to the choice of $\\lambda_m$, since monotonicity is an inherent property of LUTs.

---

> > ### Comment · Reviewer_cahF · 2025-08-05
> > **Official Comment After Rebuttal**
> >
> > I thank the authors for providing such a detailed response in the rebuttal. After reading it carefully, I found that the authors addressed most of the issues that I raised in the initial phase. For instance, the technical details of method and extral experiments. To this end, I choose to keep my score.

---

> ### Author Response · Authors · 2025-08-05
>
> Thank you for taking your valuable time to carefully read our rebuttal. We are honored that our efforts to address the issues you raised.  We sincerely thank you for your time and professional review.

---

### Official Review · Reviewer_QT6g · 2025-06-27

**Clarity:** 4
**Significance:** 4
**Originality:** 4
**Rating:** 5
**Confidence:** 5

**Summary:**

The paper introduces Pan-LUT, a look-up table (LUT) based method designed for the pan-sharpening task. Unlike contemporary deep learning-based approaches that require significant computational resources due to their complex network architectures, the proposed method is computationally efficient. The proposed framework incorporates three types of learnable LUTs that map values from low-resolution multispectral inputs to high-resolution multispectral outputs.

**Questions:**

1. The authors need to analyze the impact of different LUT sizes on inference time.
2. What is the maximum image resolution that the proposed method can handle on the specified GPU?.
3. What contributions does the rotation ensemble strategy outlined in Section 3.3 make to the performance or robustness of the approach?

**Ethical Concerns:**

["NO or VERY MINOR ethics concerns only"]

**Final Justification:**

The author has addressed all the issues I was concerned about, so I have decided to increase the score.

**Limitations:**

yes

**Quality:**

3

**Strengths And Weaknesses:**

Strengths:
1. The idea of using look-up table for efficient pan-sharpening is new. To the best of our knowledge, this work is the first study on LUT-based pan-sharpening.
2. The proposed method is fast, and it can process 4K-resolution images in under 1 ms using a single NVIDIA GeForce RTX 2080 Ti GPU.
3. The approach is simple, and its computational efficiency is outstanding.

Weaknesses:
1. The proposed method focuses on computational efficiency, which may come at the cost of achieving the highest performance.
2. Some parts of the paper lack clarity. For example, the meaning of "OOM" in Table 1 is not explained.

---

> ### Author Rebuttal · Authors · 2025-07-29
>
> Thank you for your review.
> # Weaknesses1
>
> Our Pan-LUT is specifically designed to efficiently process high-resolution remote sensing images under resource-constrained conditions, even if this focus on computational efficiency may come at the cost of achieving peak performance. Its lightweight architecture enables practical deployment, effectively bridging the gap to real-world applications.
>
> # Weaknesses2
>
> "OOM" in Table 1 refers to “Out of Memory”. We will revise the table caption and related descriptions for better clarity.
>
> # Question1
>
> We independently evaluate the inference time of each LUT module with different sizes $M$ to assess their computational efficiency, where $M$ denotes the number of bins per dimension. Additionally, we evaluate the SDLUT module equipped with the Rotation Ensemble Strategy (RES) for comparison. All experiments are conducted using input images with a resolution of $4096 \\times 4096$.
> | Module | M=3 | M=6 | M=9 | M=17 | M=33 |
> | :---: | :-: | :--: | :--: | :---: | :---: |
> || Inference (ms) | Inference (ms) | Inference (ms) | Inference (ms) | Inference (ms) |
> | AOLUT | 0.0677 | 0.1078 | 0.1247 | 0.2451 | 0.2988 |
> | SDLUT + RES | 0.2127 | 0.2569 | 0.3275 | 0.7142 | 0.9564 |
> | SDLUT | 0.0469 | 0.0685 | 0.1015 | 0.1474 | 0.2063 |
> | PGLUT | 0.1027 | 0.1179 | 0.1345 | 0.2046 | 0.2779 |
>
> From this table, we observe that the RES operation contributes to the majority of the inference time, as any operation on high-resolution images tends to be computationally expensive.
>
> # Question 2
>
> We evaluate the maximum image resolution that our model can handle on different GPUs.
>
> (e.g. '1K' denotes an image size of $1000 \\times 1000$ pixels.)
>
> | Models | 2080Ti (11G) | 3080Ti (12G) | 3090 (24G) | A40 (46G) | A800 (80G) |
> | :---: | :---: | :---: | :---: | :---: | :---: |
> ||resolution|resolution|resolution|resolution|resolution|
> | Pan-LUT-1 | 8.2K | 8.5K | 11.1K | 16.2K | 20.2K |
> | Pan-LUT-2 | 7.5K | 7.7K | 9.4K | 12.4K | 14.2K |
> | Pan-LUT-3 | 5.6K | 5.7K | 7.3K | 10.1K | 11.6K |
>
> Our method is sufficiently lightweight to process 4K to 8K resolution images on a single NVIDIA 2080Ti GPU.
>
> # Question 3
>
> We demonstrate the advantages of the Rotation Ensemble Strategy (RES) from both theoretical and empirical perspectives.
>
> **Reducing Memory Usage.**
> The RES effectively increases the receptive field of SDLUT from $2 \\times 2$ to $3 \\times 3$. Without RES, achieving the same receptive field would require increasing the LUT dimensionality from 4 to 9, leading to a memory complexity of $O(N^9)$. In contrast, RES enables a larger contextual receptive field without increasing LUT size, thereby reducing memory usage.
>
> **Local Texture Enhancement.**
> Let $(w, h)$ denote the pixel located at the w-th column and h-th row of the image.
> We provide a proof that RES extends the receptive field from $2 \\times 2$ to $3 \\times 3$.
> Ideally, we aim to incorporate all pixels within the local $3 \\times 3$ region centered at $(w, h)$, corresponding to the input index set:
>
> $G_{tgt} = \\{(w{-}1,h{-}1), (w,h{-}1), (w{+}1,h{-}1), (w{-}1,h), (w,h), (w{+}1,h), (w{-}1,h{+}1), (w,h{+}1), (w{+}1,h{+}1)\\}.$
>
> By default, SDLUT focuses on the bottom-right neighborhood of each pixel.  The input index group is defined as
>
> $G_{r0} = \\{(w,h), (w+1,h), (w,h+1), (w+1,h+1)\\}.$
>
> With the proposed **RES**, we apply SDLUT across three rotated versions of the input, effectively capturing additional pixel neighborhoods:
> $G_{r90} = \\{(w,h), (w{-}1,h), (w,h{+}1), (w{-}1,h{+}1)\\}$,
>
> $G_{r180} = \\{(w,h), (w{-}1,h), (w,h{-}1), (w{-}1,h{-}1)\\}$,
>
> $G_{r270} = \\{(w,h), (w{+}1,h), (w,h{-}1), (w{+}1,h{-}1)\\}$.
>
> Then, we can derive the following equation:
>
> $G_{r0} \\cup G_{r90} \\cup G_{r180} \\cup G_{r270} = G_{tgt}.$
>
> **Definition:** w/o RES' means the framework without the Rotation Ensemble Strategy.
> | Method | WV2 | GF2 | WV3 |
> | :---: | :---: | :---: | :---: |
> | Pan-LUT-1 | 40.6362 | 43.6559 | 29.6213 |
> | Pan-LUT-1 (w/o RES)| 40.4993 | 43.2147 | 29.3354 |
> | Pan-LUT-2| 40.8419 | 44.5545 | 29.7929 |
> | Pan-LUT-2 (w/o RES)| 40.7247 | 44.3754 | 29.6786 |
> | Pan-LUT-3| 41.1362 | 44.8559 | 30.1213 |
> | Pan-LUT-3 (w/o RES)| 40.9988 | 44.6672 | 30.0455 |
>
> We find that RES not only contributes to performance improvement but also significantly reduces the number of parameters.

---

> ### Author Response · Authors · 2025-08-07
>
> Thank you very much for your positive feedback. We are delighted to hear that our rebuttal has successfully addressed your concerns. We sincerely appreciate you increasing the score for our manuscript. Your kind consideration is a great encouragement to us. Thank you once again for your valuable time and constructive comments.

---

### Official Review · Reviewer_sEAR · 2025-06-30

**Clarity:** 3
**Significance:** 3
**Originality:** 3
**Rating:** 5
**Confidence:** 5

**Summary:**

Pan-LUT bridges the gap between computational efficiency and performance in high-resolution pan-sharpening. By replacing complex DNN operations with learnable LUTs, it achieves speedup over SOTA methods while maintaining competitive accuracy. This work pioneers hardware-friendly LUT designs for remote sensing, with implications for real-time Earth observation systems.

**Questions:**

1. This paper aims for "real-world applicability" but ignores key satellite-specific resolution variations, undermining claims like bridging the gap to real-world applications. This method has been experimentally tested with a fixed downsampling factor of 4. How to deal with situations where the downsampling factor is not fixed?

2. This method has only been validated in scenarios with a small number of spectra. How to deal with a large number of input data spectra?

3. In this paper, it is mentioned multiple times that the application of existing DL methods is limited in scenarios without GPUs and CPUs, overlooking critical edge device validation despite claiming “lightweight” and “hardware-friendly” attributes. Can comparative data be provided on edge devices?

**Ethical Concerns:**

["NO or VERY MINOR ethics concerns only"]

**Final Justification:**

The authors have addressed my concerns. I lean to keep the score.

**Limitations:**

Yes

**Paper Formatting Concerns:**

There are no major formatting issues.

**Quality:**

3

**Strengths And Weaknesses:**

Strengths:

1. This paper proposed a learnable LUT framework for pan-sharpening, breaking computational bottlenecks of traditional DNNs.

2. Pan LUT consists of three core modules: PGLUT for controlling the spectral transformation, SDLUT for capturing spatial details, and AOLUT for adaptively Aggregating channel information.

3. Pan LUT is capable of processing ultra-high resolution (such as 4k) images, which many methods cannot achieve.

Weaknesses：

1. There is still a gap compared to SOTA methods in terms of spectral fidelity and spatial detail preservation. In addition, Lightweight is one of the characteristics of the proposed method, but it has not been compared with recent lightweight methods in experiments.

2. There is no specific explanation on how SDLUT processes boundary pixels in section 3.

3. The dataset used in the experiment has a limited number of spectra, and the proposed method did not consider how to cope with the increase in the number of spectra, especially in hyperspectral scenarios.

---

> ### Author Rebuttal · Authors · 2025-07-29
>
> Thank you for your review.
> # Weakness 1
> [1] LGPConv-Learnable-Gaussian-Perturbation-Convolution-for-Lightweight-Pansharpening. IJCAI 2023
>
> [2] Deep Adaptive Pansharpening via Uncertainty-aware Image Fusion. TGRS 2023
>
> [3] SSUN-Net: Spatial-Spectral Prior-Aware Unfolding Network for Pan-Sharpening. AAAI 2025
>
> UAPN-S is the most lightweight model variant proposed in [2]. SSUN-Net-1stg is refers to the version with a single reconstruction stage in [3].
> | Method | WV2 | GF2 | WV3 | Param(M) | Inference(ms) | Inference(ms) | Inference(ms)| Inference(ms)|
> | :---: | :---: | :---: | :---: | :---: | :---: | :---: | :---: | :---: |
> |||||| $2048\\times2048$| $4096\\times4096$|$2048\\times2048$ (CPU) |$4096\\times4096$ (CPU) |
> | Pan-LUT-1 | 40.6362 | 43.6559 | 29.6213 | 0.9579 | **0.3812** | **0.5445** |**1012.15**|**5338.36**|
> | Pan-LUT-2 | 40.8419 | 44.5545 | 29.7929 | 1.3843 | **0.9434** | **1.0824** |**1945.78**|**9664.84**|
> | Pan-LUT-3 | 41.1362 | 44.8559 | 30.1213 | 1.8108 | **1.4821** | **1.9648** |**2844.32**|**12523.24**|
> | LGPConv | **41.6742** | 46.6645 | **30.4232** | **0.0262** | 27.15 | OOM |25113.29|126547.26|
> | UAPN-S | 41.5663 | **46.8212** | 30.3065 | 0.0345 | OOM | OOM |78775.02|463461.26|
> | SSUN-Net-1stg | 41.6663 | 46.6343 | 30.3371 | 0.0596 | 163.69 | OOM |34446.80|202476.98|
>
> We show that current lightweight methods are only lightweight in the context of low-resolution images. These approaches are highly sensitive to input image size and rely heavily on specialized hardware such as GPUs or TPUs. As shown in Table, in the absence of GPU acceleration, these methods require a considerable amount of time to process images. UAPN-S is particularly sensitive to input resolution and fails to process $2048 \\times 2048$ images even on a single 2080Ti GPU.
>
> # Weakness 2
>
> To handle boundary pixels, we adopt a reflect padding operation, which pads the image by mirroring adjacent pixels at the borders.
>
> # Weakness 3 and Question 2
>
> To handle a larger number of spectral channels, one can simply increase the dimensionality of PGLUT. However, this leads to a rapid growth in the number of parameters. For example, in the case of PGLUT with 12 input spectral channels and a LUT size of 3, the number of parameters reaches $12 \\times 3^{12} = 6{,}377{,}292$. Since LUTs store precomputed output values to avoid redundant computation, the resulting parameter explosion is inevitable. A practical solution is to incorporate smart architectural designs—such as the proposed Rotation Ensemble Strategy (RES)—to mitigate the memory and parameter overhead.
>
> **Rank Factorization**. In multispectral and hyperspectral scenarios, rank factorization offers a promising direction to reduce the parameter count and memory consumption of the model. We plan to explore this strategy in future research.
>
> # Question 1
>
> Typically, the LRMS images are upsampled to match the PAN resolution using bicubic interpolation before being fed into model, which is a widely adopted practice in pansharpening literature. Our method follows this approach and is inherently compatible with varying resolution ratios. Regardless of the original downsampling factor, once the LRMS image is interpolated to the PAN resolution, Pan-LUT can be applied without requiring any architectural modifications. This ensures the robustness and applicability of our method in real-world scenarios with diverse resolution settings.
>
> # Question 3
>
> Our method focuses on scenarios involving high-resolution image processing, particularly in environments where GPUs or TPUs are unavailable, as stated in Lines 6 and 50 of the manuscript. As shown in the table provided in our response to Weakness 1, our method can be efficiently deployed on CPUs while maintaining computational efficiency. In practice, many edge devices, such as the NVIDIA Jetson and Rockchip RK series, are equipped with CPUs. And our method can be deployed on any edge device equipped with a CPU.

---

> > ### Comment · Reviewer_sEAR · 2025-08-05
> >
> > I appreciate the authors' detailed responses to my concerns regarding the upsampling ratio, the lack of comparison with recent lightweight SOTA methods, and the absence of the testing results on edge devices. The clarifications have adequately addressed these points and I will keep my original score.

---

> ### Author Response · Authors · 2025-08-05
>
> Thank you very much for your continued attention and thoughtful review of our work. Your professional insights have been invaluable in enhancing the quality of our paper. We sincerely appreciate your time and constructive comments.

---

### Official Review · Reviewer_Znrg · 2025-07-02

**Clarity:** 3
**Significance:** 3
**Originality:** 4
**Rating:** 5
**Confidence:** 5

**Summary:**

In this paper, to achieve high computational efficiency and a lightweight design, the authors propose a fully LUT-based framework. Specifically, they design three types of LUT modules: PGLUT for spectral information exchange across channels, SDLUT for capturing fine-grained local spatial details, and AOLUT for aggregating multimodal information to produce adaptive high-resolution results. Experimental results demonstrate that the proposed method achieves superior computational efficiency while delivering performance comparable to state-of-the-art deep learning-based approaches.

**Questions:**

1. Pan-LUT-N denotes the model variant with N stacked PGLUT and SDLUT modules, analogous to increasing network depth. It would be valuable to evaluate its generalization ability across different resolutions by testing on full-resolution images and assessing the potential risk of overfitting.

2. The rotation ensemble strategy is typically used as a Test-Time Augmentation (TTA) technique. However, its effectiveness during training remains unclear and should be validated through ablation studies or comparisons with appropriate baselines.

3. Can quadrilinear and pentalinear interpolation be regarded as higher-dimensional extensions of bilinear interpolation?

**Ethical Concerns:**

["NO or VERY MINOR ethics concerns only"]

**Final Justification:**

The author added additional experiments to address the issues I was concerned about and provided detailed formula derivations, so I decided to increase my score.

**Limitations:**

yes

**Quality:**

3

**Strengths And Weaknesses:**

Strengths:

1. This method is capable of processing high-resolution images even in resource-constrained environments. As demonstrated in the paper, it can handle images with resolutions up to 8K and 16K.

2. The proposed method is novel, fast, and computationally efficient, with a simple yet effective design that makes it practical for real-world applications.

Weaknesses:

1. As mentioned in the paper, the storage requirements increase substantially with the size and number of LUTs, which is an inherent limitation of LUT. This reflects a trade-off between performance and computational efficiency, as well as between memory requirements and processing speed.

2. Some details in the writing are not clearly presented. The paper uses two different forms of resolution description. In Section 1, the terms "2K" and "4K" are used, while in Section 4.3 (line 269), the resolutions are specified as 2048 × 2048 and 4096 × 4096. It is unclear whether "2K" specifically refers to 2048 × 2048. This inconsistency may cause confusion and should be clarified.

---

> ### Author Rebuttal · Authors · 2025-07-25
>
> Thank you for your review.
>
> # Weakness 1
>
> There is a trade-off between performance and computational efficiency associated with LUT-based methods. However, this overhead is acceptable given the performance benefits achieved, especially in real-time inference and high-resolution image processing scenarios.
>
> # Weakness 2
>
> "2K" specifically refers to a resolution of $2048\\times2048$, while "4K" refers to $4096\\times4096$, we will revise the manuscript to clarify the actual image sizes used in our experiments to avoid ambiguity.
>
> # Question 1
>
> We conduct additional experiments to evaluate more Pan-LUT-N variants, where $N$ denotes the number of stacked PGLUT and SDLUT modules.
>
> | Method | WV2 | GF2 | WV3 | Param(M) | Inference(ms) | Inference(ms) |
> | :---: | :---: | :---: | :---: | :---: | :---: | :---: |
> ||||||$2048\\times2048$|$4096\\times4096$
> | Pan-LUT-1 | 40.6362 | 43.6559 | 29.6213 | 0.9579 | 0.3812 | 0.5445 |
> | Pan-LUT-2 | 40.8419 | 44.5545 | 29.7929 | 1.3843 | 0.9434 | 1.0824 |
> | Pan-LUT-3 | 41.1362 | 44.8559 | 30.1213 | 1.8108 | 1.4821 | 1.9648 |
> | Pan-LUT-4 | 41.1624 | 44.9756 | 30.1913 | 2.2373 | 2.0621 | 2.7518 |
> | Pan-LUT-5 | 41.2854 | 45.1733 | 30.2471 | 2.6637 | 2.3712 | 2.9915 |
> | Pan-LUT-6 | 41.4025 | 45.3364 | 30.3087 | 3.0902 | 2.7915 | 3.1024 |
>
> As shown, deeper models (e.g., Pan-LUT-4 to Pan-LUT-6) offer only marginal improvements in performance compared to shallower variants (Pan-LUT-1 to Pan-LUT-3), while significantly increasing parameter count. For example, Pan-LUT-4 already exceeds 2 million parameters, which limits its practicality. Therefore, we adopt Pan-LUT-1, 2, and 3 as the default architectural choices.
>
> | Metrics | Pan-LUT-1 | Pan-LUT-2 | Pan-LUT-3 | Pan-LUT-4 | Pan-LUT-5 | Pan-LUT-6 |
> | :-----: | :-----: | :-----: | :-----: | :-----: | :-----: | :-----: |
> | $D_{\\lambda} \\downarrow$ | 0.0956 | 0.0962 | 0.0978 | 0.0963 | 0.0987 | 0.0944 |
> | $D_{S} \\downarrow$ | 0.1145 | 0.1253 | 0.1276 | 0.1269 | 0.1212 | 0.1233 |
> | QNR$\\uparrow$ | 0.8024 | 0.7985 | 0.7948 | 0.7934 | 0.7959 | 0.7986 |
> **Dataset: Full-WV-II**
>
> Although deeper architectures often risk overfitting, the consistent performance of Pan-LUT models on unseen datasets indicates that our method maintains strong generalization without overfitting the training data.
>
> # Question 2
>
> We demonstrate the advantages of the Rotation Ensemble Strategy (RES) from both theoretical and empirical perspectives.
>
> **Reducing Memory Usage.**
> The RES effectively expands the receptive field of the SDLUT module from $2 \\times 2$ to $3 \\times 3$. Without RES, achieving the same receptive field would require increasing the LUT dimensionality from 4 to 9, leading to a memory complexity of $O(N^9)$. In contrast, RES enables a larger receptive field without increasing LUT size, thereby reducing memory usage.
>
> **Local Texture Enhancement.**
> Let $(w, h)$ denote the pixel located at the w-th column and h-th row of the image. We provide a theoretical proof that RES extends the receptive field from $2 \\times 2$ to $3 \\times 3$.
> Ideally, we aim to incorporate all pixels within the local $3 \\times 3$ region centered at $(w, h)$, corresponding to the input index set:
>
> $G_{tgt} = \\{(w{-}1,h{-}1), (w,h{-}1), (w{+}1,h{-}1), (w{-}1,h), (w,h), (w{+}1,h), (w{-}1,h{+}1), (w,h{+}1), (w{+}1,h{+}1)\\}.$
>
> By default, SDLUT captures the bottom-right neighborhood of each pixel, with the corresponding input index group defined as:
>
> $G_{r0} = \\{(w,h), (w+1,h), (w,h+1), (w+1,h+1)\\}.$
>
> With the proposed **RES**, SDLUT is applied to three rotated versions of the input, thereby effectively capturing a broader set of pixel neighborhoods.
>
> $G_{r90} = \\{(w,h), (w{-}1,h), (w,h{+}1), (w{-}1,h{+}1)\\}$,
>
> $G_{r180} = \\{(w,h), (w{-}1,h), (w,h{-}1), (w{-}1,h{-}1)\\}$,
>
> $G_{r270} = \\{(w,h), (w{+}1,h), (w,h{-}1), (w{+}1,h{-}1)\\}$.
>
> Then, we can derive the following equation:
>
> $G_{r0} \\cup G_{r90} \\cup G_{r180} \\cup G_{r270} = G_{tgt}.$
>
>
> | Method | WV2 | GF2 | WV3 |
> | :---: | :---: | :---: | :---: |
> | Pan-LUT-1 | 40.6362 | 43.6559 | 29.6213 |
> | Pan-LUT-1 (w/o RES)| 40.4993 | 43.2147 | 29.3354 |
> | Pan-LUT-2| 40.8419 | 44.5545 | 29.7929 |
> | Pan-LUT-2 (w/o RES)| 40.7247 | 44.3754 | 29.6786 |
> | Pan-LUT-3| 41.1362 | 44.8559 | 30.1213 |
> | Pan-LUT-3 (w/o RES)| 40.9988 | 44.6672 | 30.0455 |
>
> **Definition:** w/o RES' means the framework without the Rotation Ensemble Strategy.
>
> We find that RES not only contributes to performance improvement but also significantly reduces the number of parameters.
>
> # Question 3
> The proposed quadrilinear and pentalinear interpolation methods are conceptually similar to linear and bilinear interpolation, serving as their extensions to higher dimensions. Below, we provide a theoretical analysis to illustrate their similarities.
>
> **Linear Interpolation.**
> Given two known data points $P_0=(x_0,y_0)$ and $P_1=(x_1,y_1)$, we assume the function changes linearly between these two points. We then estimate the corresponding $y$ value for a target point $x$ based on its proportional position between $x_0$ and $x_1$.
> Specifically, if a point $x$ is between $x_0$ and $x_1$, and its distance to $x_0$ is $t$ times the total distance $|x_1−x_0|$ (where $0≤t≤1$), then $x=x_0+t(x_1−x_0)$.
> Similarly, its corresponding $y$ value, $y$, can be expressed as $y=(1-t)y_0+ty_1$.
>
> **Bilinear Interpolation.**
> Suppose we are given function values $f$ at four corner points on a 2D plane: $Q_{11}=(x_1,y_1)$, $Q_{12}=(x_1,y_2)$, $Q_{21}=(x_2,y_1)$, and $Q_{22}=(x_2,y_2)$. We aim to estimate the function value $f(x,y)$ for any point $P=(x,y)$ on the plane.
>
> **Step 1: Linear interpolation along the x-direction.**
>
> First, along the row where $y_1$ is constant, linearly interpolate the point $(x, y_1)$. Using the values of $Q_{11}=(x_1, y_1)$ and $Q_{21}=(x_2, y_1)$, calculate the value for $R_1=(x, y_1)$:
>
> $f(R_1) = \\frac{x_2 - x}{x_2 - x_1} f(Q_{11}) + \\frac{x - x_1}{x_2 - x_1} f(Q_{21})=d_xf(Q_{11})+(1-d_x)f(Q_{21})$,
>
> where $d_x = \\frac{x_2 - x}{x_2 - x_1}$ denotes the relative distance to the known points.
>
> Then, along the row where $y_2$ is constant, linearly interpolate the point $(x, y_2)$. Using the values of $Q_{12}=(x_1, y_2)$ and $Q_{22}=(x_2, y_2)$, calculate the value for $R_2=(x, y_2)$:
>
> $f(R_2) = \\frac{x_2 - x}{x_2 - x_1} f(Q_{12}) + \\frac{x - x_1}{x_2 - x_1} f(Q_{22})=d_xf(Q_{12})+(1-d_x)f(Q_{22}).$
>
> **Step 2: Linear interpolation along the y-direction.**
>
> Finally, using the values of $R_1=(x, y_1)$ and $R_2=(x, y_2)$, linearly interpolate for the target point $P=(x, y)$:
>
> $f(P) = \\frac{y_2 - y}{y_2 - y_1} f(R_1) + \\frac{y - y_1}{y_2 - y_1} f(R_2)=d_xd_yf(Q_{11})+(1-d_x)d_yf(Q_{21})+d_x(1-d_y)f(Q_{12})+(1-d_x)(1-d_y)f(Q_{22}),$
>
> where $d_y =  \\frac{y_2 - y}{y_2 - y_1}$ denotes the relative distance to the known points.
>
> **Quadrilinear Interpolation in Look up table.**
> All the interpolation methods above exhibit a structurally similar form in estimating the target value:
>
> $y=(1-t)y_0+ty_1$,
>
> $f(P) = d_xd_yf(Q_{11})+(1-d_x)d_yf(Q_{21})+d_x(1-d_y)f(Q_{12})+(1-d_x)(1-d_y)f(Q_{22})$.
>
> We show that the estimation formula of quadrilinear interpolation follows the same functional form.Since quadrilinear interpolation involves 16 known points, the full expression is too lengthy to present here. For more details, please refer to the technical details section in the supplementary material.

---

> > ### Comment · Reviewer_Znrg · 2025-08-05
> >
> > The author's careful explanation resolved my doubts about the technical details, and I decided to adjust the score I gave.

---

> ### Author Response · Authors · 2025-08-05
>
> Thank you for your response and feedback. We are glad that our explanation could resolve your questions. This is a great encouragement to us. Thank you again for your valuable time and professional review.

---

### Decision · Program_Chairs · 2025-09-17

**Decision:**

Accept (oral)

**Comment:**

This paper studies the problem of efficient pan-sharpening in remote sensing. By replacing complex DNN operations with learnable LUTs,  Pan-LUT bridges the gap between computational efficiency and performance in high-resolution pan-sharpening.

All reviewers recognize the novelty of using a learnable LUT framework and its impact to remote sensing. Compared to previous approaches it enables processing of high-res images in practice, which is a valuable contribution to the field of remote sensing image processing.